# Screening and Characterization of TAT-Fused Nanobodies Targeting Bovine Viral Diarrhea Virus NS3/NS5A for Antiviral Application

**DOI:** 10.3390/biom15111593

**Published:** 2025-11-13

**Authors:** Qianqian Dong, Yangyang Xiao, Zhao Liu, Wenxiang Zhang, Aodi Wu, Hanwen Zhang, Jinliang Sheng

**Affiliations:** College of Animal Science and Technology, Shihezi University, Shihezi 832003, China; dongqianqian@stu.shzu.edu.cn (Q.D.); shzuxyy@163.com (Y.X.); liuzhao09192025@163.com (Z.L.); z302729814@163.com (W.Z.); 15739592133@163.com (A.W.); aa747582693@163.com (H.Z.)

**Keywords:** bovine viral diarrhea virus (BVDV), nanobody, trans-activator of transcription (TAT), antiviral

## Abstract

Bovine viral diarrhea virus (BVDV) is a major pathogen responsible for significant economic losses in the global cattle industry. The diverse transmission routes and the characteristics of asymptomatic infections make it difficult to contain the spread; there is an urgent need to develop new effective antiviral strategies. Nanobodies (Nbs) have become a promising new type of antiviral agent due to their advantages, including small molecular size, stable structure, high specificity, and ease of production. This study successfully screened a specific nanobody, Nb7, targeting the key functional protein NS5A of BVDV using phage display technology. Furthermore, the nanobody was effectively delivered into Madin–Darby bovine kidney (MDBK) cells by fusing it with the cell-penetrating peptide TAT. The results demonstrate that TAT-Nb7, specifically targeting the non-structural protein NS5A of BVDV, significantly inhibits viral replication in MDBK cells. In conclusion, this study indicates that TAT-Nb7 holds promise as a therapeutic candidate for the prevention and control of BVDV infection.

## 1. Introduction

Bovine viral diarrhea virus (BVDV), a major pathogen causing substantial economic losses in the global cattle industry, presents a wide range of clinical manifestations, from subclinical infection to diarrhea, fever, pneumonia, hemorrhagic lesions, and even fatal mucosal disease (MD) [1]. Cattle of all ages are susceptible to BVDV, and the virus has diverse transmission routes. Horizontal transmission can easily lead to acute infections [2], while infections during pregnancy typically result in vertical transmission of the virus from the mother to the developing fetus [3]. Of particular importance, vertical transmission can give rise to PI calves, which remain persistently infected and continuously shed the virus throughout their lives, representing the most dangerous source of infection within herds [4]. In addition to cattle, BVDV can infect a variety of domestic and wild species [5]. Beyond its clinical significance, BVDV is also a major contaminant in laboratory settings. Numerous commercial sera [6], cell lines, and vaccines [7] have been reported to be contaminated with this virus. Therefore, BVDV has become a central focus of global control and eradication programs [8].

BVDV belongs to the genus Pestivirus of the family Flaviviridae. Its genome is a single-stranded positive-sense RNA molecule that encodes a single polyprotein, cleaved into 11 mature proteins: four structural proteins (C, Erns, E1, and E2) and seven non-structural proteins (Npro, NS2, NS3, NS4A, NS4B, NS5A, NS5B) [9]. Among these, NS3 is a core component of the BVDV replication complex and possesses multiple enzymatic activities [10]. As a serine protease, NS3 forms a complex with NS4A and participates in the cleavage of the viral polyprotein [11]. In addition, NS3 exhibits helicase and nucleoside triphosphatase (NTPase) activities, which are essential for unwinding viral RNA and providing energy during RNA replication, respectively [12]. NS5A, a highly phosphorylated hydrophilic protein [13], plays critical roles in viral RNA replication and immune evasion [14]. NS5A regulates RNA replication through its interaction with NS5B and the 3′-UTR. Notably, even in the absence of NS5B binding, NS5A can still modulate viral RNA synthesis by interacting with the 3′-UTR [15]. The phosphorylation status of NS5A is crucial for its function, influencing the initiation, elongation efficiency, and overall level of viral RNA replication [16]. Moreover, NS5A interferes with host cell signaling pathways, suppresses the host immune response, and contributes to persistent viral infection [17]. Both NS3 and NS5A are indispensable for the BVDV replication cycle and are relatively conserved [18], making them ideal targets for antiviral interventions.

Although vaccination has been utilized to some extent for the prevention and control of BVDV, several limitations persist, including safety concerns, insufficient immunogenicity, and the occurrence of immune tolerance. Furthermore, there is currently no effective therapeutic agent available for the treatment of BVDV infection [19], underscoring the urgent need for the development of novel and effective therapeutic strategies. Nanobodies (Nbs), as an emerging form of antibodies, have opened new avenues for BVDV research and control. These single-domain antibodies are derived from the variable domain of heavy-chain-only antibodies (VHHs) naturally present in camelids [20]. Compared to conventional antibodies, nanobodies offer several distinct advantages: they are small in molecular size [21], highly stable, and capable of retaining structural and functional integrity under extreme conditions such as high temperatures and extreme pH environments [22]. Moreover, nanobodies exhibit strong antigen specificity and can recognize epitopes that are often inaccessible to conventional antibodies [23]. To date, nanobodies have demonstrated significant potential in antiviral applications [24], laying a solid foundation for their exploration in the treatment of BVDV.

However, the ability of nanobodies to autonomously enter cells is limited by the physical barrier of the cell membrane and the complexity of endocytic pathways [25], which to some extent restricts their functional efficacy. TAT peptide, a classical cell-penetrating peptide (CPP), is derived from the amino acid sequence spanning the 48th to 57th residues of the Trans-Activator of Transcription (Tat) protein from human immunodeficiency virus type 1 (HIV-1). TAT exhibits a remarkable capacity to traverse cell membranes and can efficiently deliver a variety of biomolecules into cells while preserving their biological activity [26]. Recently, TAT-conjugated nanobodies have been employed in antiviral strategies [24]. Through this tandem design, the TAT peptide facilitates the cellular entry of nanobodies targeting non-structural proteins, thereby contributing to improved antiviral activity.

In this study, alpacas were immunized with recombinant NS3 and NS5A proteins to construct a VHH gene library. Phage display technology was used to screen specific nanobodies targeting NS3 and NS5A. Subsequently, the selected nanobodies were fused with the TAT cell-penetrating peptide to generate TAT-Nbs. Compared with the existing literature, this is the first strategy to screen specific nanobodies targeting BVDV NS5A and NS3 and combine them with the TAT delivery system to validate antiviral efficacy. These fusion proteins were expressed using a prokaryotic *E. coli* expression system, and their ability to penetrate cell membranes and enter cells was confirmed. Furthermore, TAT-Nbs showed an inhibitory effect on BVDV replication in MDBK cells, providing a potential strategy for developing novel antiviral approaches against BVDV.

## 2. Materials and Methods

### 2.1. Cell Line and Virus

The MDBK cell line was kindly provided by Professor Xiaomin Zhao from the College of Veterinary Medicine, Northwest A&F University, and has been maintained in our laboratory since then. The detailed procedure is as follows: when MDBK cells cultured in T75 flasks reached approximately 70% confluence, the culture medium was replaced with Dulbecco’s Modified Eagle Medium (DMEM). The cells were then inoculated with BVDV suspension at a multiplicity of infection (MOI) of 0.1, followed by incubation in a 37 °C, 5% CO_2_ incubator for 2 h. After incubation, the medium was removed, and the cell monolayer was rinsed twice with phosphate-buffered saline (PBS). Subsequently, DMEM supplemented with 2% fetal bovine serum (FBS) was added to the flasks, and the cells were further cultured for 3–5 days until extensive cytopathic effects (CPEs) were observed. The cells were subjected to three cycles. The virus-containing supernatant was collected by centrifugation at 8000 rpm for 10 min to remove the precipitate. After sterilization through a 0.22 μM filter, the viral supernatant was stored at −80 °C.

### 2.2. Expression and Purification of BVDV NS3 and NS5A Recombinant Proteins

The genes encoding the NS3 (GenBank accession no. NP_776267.1) and NS5A gene (GenBank accession no. NP_776270.1) proteins were cloned into the pET-30a vector by China Detai Bioengineering Co., Ltd. (Nanjing, China), to construct the recombinant plasmids pET-30a-NS3 and pET-30a-NS5A. The recombinant plasmids were then transformed into *E. coli* BL21 (DE3) competent cells. Single colonies were picked and inoculated into Terrific Broth (TB) medium supplemented with kanamycin (KAN, 50 μg/mL), followed by cultivation at 37 °C with shaking at 200 rpm for large-scale culture. When the optical density at 600 nm (OD_600_) reached 0.6, protein expression was induced with 0.2 mM isopropyl-β-D-thiogalactopyranoside (IPTG) at 15 °C for 12 h. Expression of the NS3 and NS5A proteins was confirmed by sodium dodecyl sulfate-polyacrylamide gel electrophoresis (SDS-PAGE). Subsequently, the recombinant NS3 and NS5A proteins were purified using nickel-nitrilotriacetic acid (Ni-NTA) resin, and the purity was verified by SDS-PAGE and Western blotting. Finally, the purified recombinant proteins were aliquoted and stored at −80 °C for subsequent experiments.

### 2.3. Alpaca Immunization and Construction of the VHH Phage Display Library

A healthy, pathogen-free adult alpaca was subcutaneously immunized at multiple sites with a mixture of NS3 and NS5A recombinant proteins (2 mg of each protein) emulsified in complete Freund’s adjuvant. The animal was housed at the College of Animal Science and Technology, Shihezi University, under controlled conditions with unrestricted access to food and water. Booster immunizations were administered every 14 days using the same antigens emulsified in incomplete Freund’s adjuvant, for a total of four immunizations. Blood samples were collected prior to each immunization, and serum was separated for subsequent antibody titer analysis. All procedures were performed by trained personnel to ensure animal welfare and minimize discomfort.

The serum antibody titer was assessed by indirect enzyme-linked immunosorbent assay (i-ELISA) following the fourth immunization. Briefly, 96-well microtiter plates were coated with NS3 or NS5A protein at a final concentration of 10 μg/mL. Serial dilutions of the alpaca serum were added as primary antibodies, followed by incubation with horseradish peroxidase (HRP)-conjugated goat anti-alpaca IgG (1:80,000 dilution; Abcam, Cambridge, UK) as the secondary antibody. The absorbance at 450 nm (OD_450_) was measured using a microplate reader, and the data were analyzed using GraphPad Prism 10.1.2.

After the final immunization, 80 mL of whole blood was collected for lymphocyte isolation. Peripheral blood mononuclear cells (PBMCs) were separated using a camel lymphocyte separation kit (Solarbio, Beijing, China). Total RNA was extracted from the isolated lymphocytes using an RNA extraction kit (TransGen Biotech, Beijing, China), according to the manufacturer’s instructions. First-strand cDNA was synthesized using the SuperScript™ III First-Strand Synthesis System (Thermo Fisher Scientific, Waltham, MA, USA).

The variable domain of VHH was amplified via nested PCR. The first-round PCR was performed using primers AIPVh-LD and CH2-R, yielding an approximately fragment. This product was purified and used as the template for the second-round PCR with primers VHH-Forward and VHH-Reverse, generating an approximately VHH fragment. The amplified VHH fragments were subsequently ligated into the phage display vector pCANTAB-5E to construct a VHH phage display library. All primers were synthesized by Beijing Liuhe BGI Genomics Technology Co., Ltd., Beijing, China, and the primer sequences are detailed in Table 1.

### 2.4. Preparation and Titration of Helper Phage

An overnight culture of *E. coli* TG1 harboring the recombinant pCANTAB 5E-VHH plasmid was inoculated into 2 × YT/AMP medium and incubated at 37 °C with shaking at 180 rpm for 4–5 h until the OD_600_ reached approximately 0.6. M13KO7 helper phage was then added to the bacterial culture, and the mixture was incubated statically at 37 °C for 30 min. The cells were harvested by centrifugation at 2800× *g* for 15 min, resuspended in fresh medium, and subsequently cultured overnight in 2 × YT/AMP-KAN medium.

The following day, phage particles were collected from the culture supernatant by centrifugation at 3600× *g* for 30 min at 4 °C. The supernatant was mixed with a PEG/NaCl solution and incubated on ice for 2 h. Phage particles were then precipitated by centrifugation at 12,000× *g* for 30 min. The pellet was resuspended in 1 mL of PBS and incubated overnight at 4 °C with gentle agitation (90 rpm) to ensure complete resuspension.

On the subsequent day, 100 μL of the resuspended phage solution was serially diluted 10-fold in 2 × YT medium (resulting in dilutions ranging from 10^−1^ to 10^−16^), and each dilution was incubated statically at 37 °C for 30 min. Subsequently, 100 μL of each dilution was plated onto LB agar plates supplemented with 100 μg/mL ampicillin. The plates were incubated overnight at 37 °C. The number of colonies formed at each dilution was recorded, and the phage titer was calculated using the following formula:$$Phage\;titer\;(pfu/mL)\;=\;\frac{\left(Number\;of\;plaques\;\times \;Dilution\;factor\right)}{Volume\;of\;phage\;dilution\;added\;to\;the\;plate\;(mL)}$$

### 2.5. Screening and Identification of NS3- and NS5A-Specific Nanobodies

NS3 and NS5A proteins were diluted to 10 μg/mL in PBS and used to coat a 96-well plate at 100 μL per well, followed by overnight incubation at 4 °C. The next day, the plate was incubated with 5% (*w*/*v*) skim milk at 37 °C for 2 h to block non-specific binding sites. Phage particles were resuspended in blocking buffer and diluted to a final concentration of 4.4 × 10^12^ pfu/mL, followed by incubation at 37 °C for 2 h. Unbound phages were removed by washing, and specifically bound phages were eluted with 100 μL of 0.1 M triethylamine per well at 37 °C for 10 min. The eluted phage suspension was collected into a 1.5 mL microcentrifuge tube and neutralized with 1 M Tris-HCl (pH 7.4). Subsequently, 400 μL of the neutralized phage solution was added to 4 mL of exponentially growing *E. coli* TG1 cells and incubated at 37 °C for 30 min without shaking. After infection, 16 mL of 2 × YT medium supplemented with ampicillin (AMP) was added, and the culture was grown at 37 °C until the OD_600_ reached approximately 0.6. M13KO7 helper phage was then introduced to rescue the phagemid-containing cells, and three rounds of biopanning were carried out using the same protocol to progressively enrich for antigen-specific phage clones. The efficiency of enrichment in each round was assessed by phage titration, ELISA, and colony PCR.

### 2.6. Expression of TAT-Nb Recombinant Proteins

The genes encoding Nb23 (targeting the NS3 protein) and Nb7 (targeting the NS5A protein) were fused with the cell-penetrating peptide TAT gene via a (G_4_S)_3_ linker by Zhongding Biotechnology Co., Ltd. (Nanjing, China), yielding the TAT-Nb23 and TAT-Nb7 gene sequences. These sequences were then cloned into the pET28a-sumo vector to construct the recombinant plasmids pET28a-sumo-TAT-Nb23 and pET28a-sumo-TAT-Nb7.

The recombinant plasmids were transformed into *E. coli* BL21 (DE3) competent cells. Single-colony clones after transformation were picked and inoculated into TB/KAN medium for large-scale culture, which was continued until the OD_600_ reached approximately 0.6. Protein expression was induced with 0.2 mM IPTG at 15 °C for 12 h. The expression of TAT-Nb7 and TAT-Nb23 was verified by SDS-PAGE.

Subsequently, the TAT-Nb7 and TAT-Nb23 were purified using Ni-NTA resin. The purified proteins were identified by SDS-PAGE and Western blotting, and then stored at −80 °C for subsequent use.

### 2.7. Cell Viability Assay

Confluent MDBK cells were seeded into 96-well plates at a density of 1 × 10^4^ cells per well and incubated at 37 °C in a 5% CO_2_ incubator for 12 h. Once a confluent monolayer had formed, the culture medium was removed, and the cells were washed twice with PBS. Subsequently, serum-free medium containing various concentrations of TAT-Nb7 and TAT-Nb23 (5, 10, 15, and 20 μM) was added to the wells. Each concentration was tested in eight replicate wells, and untreated normal MDBK cells served as the blank control. After incubation for 24 h at 37 °C in a 5% CO_2_ incubator, the medium was replaced with fresh serum-free medium, and 10 μL of CCK-8 reagent was added to each well. The cells were further incubated under the same conditions for 2 h, after which the OD_450_ was measured using a microplate reader. Cell viability was calculated using the following formula (as recommended by the CCK-8 kit manufacturer):Cellviability(%)=[A(with drug)−A(blank)]/[A(withou tdrug)−A(blank)]×100%.

*A* (*with drug*): Absorbance of wells containing cells, CCK-8 solution, and the test drug.

*A* (*blank*): Absorbance of wells containing culture medium and CCK-8 solution, but no cells.

*A* (*without drug*): Absorbance of wells containing cells and CCK-8 solution, but no test drug.

### 2.8. IFA Membrane Permeabilization Assay

MDBK cells were seeded into 12-well plates and cultured until they reached approximately 60% confluence. The medium was then replaced with serum-free DMEM supplemented with either 10 μM TAT-Nb7 or 10 μM TAT-Nb23, and the cells were incubated for 6 h at 37 °C in a humidified atmosphere containing 5% CO_2_. Following treatment, the cells were fixed at room temperature with 4% paraformaldehyde for 15 min, permeabilized with 0.2% Triton X-100 for 20 min, and subsequently blocked with 2% bovine serum albumin (BSA) for 1 h, all at room temperature. The samples were then incubated overnight at 4 °C with a SUMO antibody (1:1000 dilution; ZSGB-BIO, Beijing, China). After thorough washing with PBS, the cells were incubated with an Alexa Fluor 488-conjugated goat anti-mouse IgG secondary antibody (1:800 dilution; Proteintech, Wuhan, China) for 1 h at room temperature in the dark. Nuclei were counterstained with DAPI, and the samples were visualized using a SOPTOP inverted fluorescence microscope.

### 2.9. Western Blot Membrane Permeabilization Assay

MDBK cells were seeded into 6-well plates and cultured until they reached approximately 60% confluence. The medium was then replaced with serum-free DMEM supplemented with either 10 μM TAT-Nb7 or 10 μM TAT-Nb23, and the cells were incubated for 6 h at 37 °C in a humidified incubator containing 5% CO_2_. Following treatment, the cells were lysed using RIPA lysis buffer supplemented with a protease inhibitor (PMSF). Protein concentration was quantified using a BCA Protein Assay Kit (Beyotime, Shanghai, China) and normalized to 1 mg/mL.

Equal amounts of protein (20 μg per sample) were mixed with loading buffer, boiled for 5 min, and separated by SDS-PAGE. The proteins were subsequently transferred onto PVDF membranes. Membranes were blocked overnight at 4 °C with 5% non-fat milk dissolved in Tris-buffered saline containing 0.1% Tween-20 (TBST). After blocking, the membranes were incubated with primary antibodies diluted in blocking buffer: mouse anti-SUMO antibody (1:1000; ZSGB-BIO, Beijing, China) and mouse anti-GAPDH antibody (1:2000; ZSGB-BIO, Beijing, China). After washing with TBST, the membranes were incubated with HRP-conjugated goat anti-mouse IgG secondary antibody (1:5000; Abmart, Shanghai, China). Finally, the immunoreactive bands were visualized using an ECL chemiluminescence detection system, and the relative intensities of SUMO-tagged TAT-Nb bands (TAT-Nb7 and TAT-Nb23) were quantified using ImageJ v1.49 software (National Institutes of Health, Bethesda, MD, USA), with GAPDH as the internal reference for normalization.

### 2.10. RT-qPCR Neutralization Assay

MDBK cells were seeded into 6-well plates and cultured until they reached approximately 70% confluence. The cells were then inoculated with BVDV at an MOI of 1. After 2 h of viral adsorption, the infection medium was replaced with fresh culture medium containing 2% FBS supplemented with 10 μM TAT-Nb7 or 10 μM TAT-Nb23. Cells were collected at 0 h, 12 h, 24 h, 36 h, and 48 h post treatment using TRIzol reagent. Total RNA was extracted using the TransGen Biotech RNA extraction kit (Beijing, China), and cDNA was synthesized using the Takara PrimeScript RT Master Mix (Takara, Japan). Quantitative real-time PCR (qPCR) was carried out with specific primers targeting the BVDV 5′ untranslated region (UTR). The β-actin gene was used as an internal reference to normalize the relative mRNA expression levels. Gene expression was calculated using the $2^{-∆∆t}$ method. The primer sequences used for RT-qPCR are listed in Table 2.

### 2.11. Western Blot Assay for Evaluating BVDV Replication

MDBK cells were seeded into 6-well plates and cultured until they reached approximately 70% confluence. The cells were then inoculated with BVDV at an MOI of 1. After 2 h of viral adsorption, the infection medium was replaced with fresh culture medium containing 2% FBS supplemented with 10 μM TAT-Nbs, and the cells were incubated for 48 h at 37 °C in a humidified incubator containing 5% CO_2_. Following treatment, the cells were lysed using RIPA lysis buffer supplemented with a protease inhibitor (PMSF). Protein concentration was quantified using a BCA Protein Assay Kit (Beyotime, Shanghai, China) and normalized to 1 mg/mL. Equal amounts of protein (20 μg per sample) were mixed with loading buffer, boiled for 5 min, and separated by SDS-PAGE. The proteins were subsequently transferred onto PVDF membranes. Membranes were blocked overnight at 4 °C with 5% non-fat milk dissolved in Tris-buffered saline containing 0.1% Tween-20 (TBST). After membrane transfer, the PVDF membrane was cut into two strips according to the molecular weight ranges of the target protein (BVDV E2) and internal reference protein (GAPDH) to avoid cross-reactivity between primary and secondary Abs of different species. The two membrane strips were then blocked overnight at 4 °C with 5% non-fat milk dissolved in TBST. Thereafter, the membrane strip for BVDV E2 detection was incubated with rabbit anti-BVDV E2 pAb (1:1000 dilution; Biosynthesis Biotechnology Co., Ltd., Beijing, China) diluted in blocking buffer, while the membrane strip for GAPDH detection was incubated with mouse anti-GAPDH mAb (1:2000 dilution; ZSGB-BIO, Beijing, China) diluted in blocking buffer. After thorough washing with TBST, the BVDV E2-targeted membrane strip was incubated with HRP-conjugated goat anti-rabbit IgG secondary Ab (1:5000 dilution; Abmart, Shanghai, China), and the GAPDH-targeted membrane strip was incubated with HRP-conjugated goat anti-mouse IgG secondary Ab (1:5000 dilution; Abmart, Shanghai, China). Finally, the immunoreactive bands were visualized using an ECL chemiluminescence detection system, and the relative intensities of BVDV E2 bands were quantified using ImageJ software (National Institutes of Health, Bethesda, MD, USA), with GAPDH as the internal reference for normalization.

### 2.12. Molecular Docking of Nanobodies with Their Target Antigen Proteins

The amino acid sequence of the NS5A antigen protein (GenBank accession no. NP_776270.1) and the sequence of Nb7 were input into the official AlphaFold3 (AF3) platform—a state-of-the-art AI tool developed by Google DeepMind for high-precision prediction of biomolecular complex structures, whose core methodology was published in Nature in 2024 [27] (using default prediction parameters) to predict the structure of the protein complex formed by the two molecules. After the prediction process was completed, the compressed package containing the prediction results was downloaded from the platform. Upon decompression, the high-confidence PDB format file was selected. PDBE Pisa (https://www.ebi.ac.uk/pdbe/pisa/, accessed on 1 October 2025) was utilized to analyze the information, and PyMOL 3.1 software was employed for three-dimensional structural visualization analysis.

### 2.13. Statistical Analysis

Data are expressed as the mean ± SD of three independent experiments. *p* values were determined by one-way analysis of variance (ANOVA). * *p* < 0.05, ** *p* < 0.01, and *** *p* < 0.001; ns, not significant. After one-way ANOVA was used to assess overall differences among multiple groups, Dunnett’s test was further applied as the test to compare the mean values of each experimental group with the blank control group, thereby identifying specific group differences.

## 3. Results

### 3.1. Expression and Purification of BVDV NS3 and NS5A Recombinant Proteins

The BVDV NS3 and NS5A recombinant proteins were induced with IPTG and purified using Ni-NTA affinity chromatography. As shown in Figure 1A,B, SDS-PAGE analysis confirmed the successful expression of the NS3 and NS5A proteins, which had the expected molecular weights of approximately 80 kDa and 55 kDa, respectively. Western blotting with a His-tag monoclonal antibody confirmed the identity of the purified proteins, revealing distinct bands at the expected molecular weights of 80 kDa and 55 kDa (Figure 1C,D).

### 3.2. Construction and Characterization of VHH Phage Display Library

To generate a VHH phage display library, healthy adult alpacas were immunized with NS3 and NS5A recombinant proteins. A total of four immunizations were performed, with each dose containing 2.0 mg of antigen. Four days after the final immunization, serum antibody titers were evaluated using i-ELISA. The results showed that the titers of anti-NS3 and anti-NS5A antibodies reached 1:512,000, indicating a strong immune response (Figure 2A,B). Peripheral blood lymphocytes (PBLs) were isolated from whole blood, and total RNA was extracted and reverse-transcribed into cDNA for use as a PCR template.

First-round PCR amplification yielded a DNA fragment of approximately 700 bp (Figure 2C), which was gel-purified and subjected to a second round of PCR. The second round of PCR produced a ~400 bp fragment corresponding to the VHH region (Figure 2D). The purified VHH fragments were ligated into the pCANTAB-5E phagemid vector, and successful ligation was confirmed by restriction enzyme digestion (Figure 2E). The recombinant plasmids were then electroporated into *E. coli* TG1 competent cells to construct the phage display library.

The library was characterized in terms of titer, insert size, insertion efficiency, and sequence diversity. The library titer was determined to be approximately 5.2 × 10^11^ pfu/mL (Figure 2F), and the insertion rate was estimated to be ~91% based on PCR screening of 48 randomly selected clones (Figure 2G).

### 3.3. Screening and Identification of Specific Nanobodies

Using the purified target proteins (NS3 and NS5A) as coating antigens, three rounds of biopanning were performed. After these three rounds, a significant enrichment of specific phages against the NS3 and NS5A target proteins was observed. Additionally, following the three selection rounds, the recovery rate of recombinant phages gradually increased, and the enrichment factor was significantly enhanced (Table 3 and Table 4).

From the elution products of the third round of affinity selection, 96 individual monoclonal colonies were randomly selected for identification by i-ELISA (Figure 3A,B). The positive colonies were subjected to sequencing, and finally, amino acid sequence alignment was conducted based on the complementarity-determining region 3 (CDR3) of the nanobodies. As shown in Figure 3C, a total of seven sequences were identified—these sequences exhibited high homology to variable domains of VHHs and possessed distinct amino acid compositions. These nanobodies were designated as Nb7, Nb10, Nb11, Nb15, Nb19, Nb22, and Nb23.

Among them, Nb7, Nb10, Nb11, and Nb15 showed good specificity toward the NS5A protein, while Nb19, Nb22, and Nb23 exhibited excellent specificity for the NS3 protein. The nanobodies Nb7 and Nb23, which had the highest OD_450_ values, were selected for subsequent experiments (Figure 3D,E).

### 3.4. Expression and Purification of TAT-Nb7 and TAT-Nb23

The recombinant proteins pET28a-Sumo-TAT-Nb7 and pET28a-Sumo-TAT-Nb23 were expressed in *E. coli* following IPTG induction and subsequently purified using Ni-NTA affinity chromatography, followed by dialysis. SDS-PAGE (Figure 4A,B) and Western blotting (Figure 4C,D) confirmed successful expression and purification of both nanobodies. The observed molecular weights of the purified proteins were consistent with the expected values: approximately 30.16 kDa for TAT-Nb7 and 30.46 kDa for TAT-Nb23.

### 3.5. Cytotoxicity Assay and Transmembrane Verification

The viability of MDBK cells treated with recombinant TAT-Nb7 and TAT-Nb23 proteins at various concentrations was assessed using the CCK-8 assay. As shown in Figure 5A, when the concentrations of TAT-Nb7 and TAT-Nb23 were below 10 μM, no significant difference in cell viability was observed compared to the untreated group, and MDBK cells retained normal viability. Therefore, a concentration of 10 μM was selected for subsequent experiments. To evaluate the transmembrane efficiency of the prokaryote-expressed nanobodies fused with tandem cell-penetrating peptides, MDBK cells were treated with TAT-Nb7 and TAT-Nb23 at a final concentration of 10 μM, respectively. After 6 h of treatment, Western Blotting (Figure 5B) and immunofluorescence assay (IFA) (Figure 5C) confirmed that both TAT-Nb7 and TAT-Nb23 could efficiently cross the cell membrane. The results of grayscale quantitative analysis of protein bands are presented in Appendix A.

### 3.6. Evaluation of TAT-Nb7 and TAT-Nb23 on BVDV Replication in MDBK Cells

To evaluate the effect of TAT-Nb7 and TAT-Nb23 on BVDV replication in MDBK cells, the cells were inoculated with BVDV at an MOI of 1. After 2 h of viral adsorption, the cells were washed twice with PBS. The control group was then cultured in DMEM supplemented with 2% FBS, while the experimental groups were treated with TAT-Nb7, TAT-Nb23, and a non-specific control nanobody targeting PRRSV, each at a final concentration of 10 μM in DMEM containing 2% FBS. Samples were collected at 0, 12, 24, 36, and 48 h post treatment. Total RNA was extracted, reverse-transcribed into cDNA, and analyzed by RT-qPCR. As shown in Figure 6A, treatment of BVDV-infected MDBK cells with TAT-Nb7 significantly reduced the viral copy number at 12–48 h post treatment, indicating that TAT-Nb7 effectively inhibits BVDV replication during this time period. In contrast, no significant reduction in viral copy number was observed in MDBK cells treated with TAT-Nb23 compared to the BVDV-infected control group (without nanobody treatment) over the 0–48 h period (Figure 6B). To further evaluate the effect of TAT-Nb7 on BVDV replication in MDBK cells, the cells were inoculated with BVDV at an MOI of 1. After 2 h of viral adsorption, the cells were washed twice with PBS. Subsequently, the control group was cultured in DMEM supplemented with 2% FBS, while the experimental groups were treated with TAT-Nb7 or a non-specific control nanobody targeting PRRSV, with a final concentration of 10 μM in DMEM containing 2% FBS for each group. Samples were collected at 48 h post treatment. Proteins were extracted and analyzed by WB. As shown in Figure 6C, treatment of BVDV-infected MDBK cells with TAT-Nb7 significantly reduced viral protein expression at 48 h post treatment, further demonstrating that TAT-Nb7 effectively inhibits BVDV replication. The results of grayscale quantitative analysis of protein bands are presented in Appendix A.

### 3.7. Molecular Docking Analysis of TAT-Nb7 with NS5A

Molecular docking (Figure 7) analysis revealed that the nanobody TAT-Nb7 forms a stable binding conformation with the NS5A antigen protein, with a binding free energy of −4.1 kcal/mol, indicating a certain binding affinity. This interaction is further stabilized by multiple synergistic non-covalent interactions at the binding interface, including ten hydrogen bonds (e.g., the relatively shorter bonds between ILE-103 and ALA-112, as well as between SER-54 and ARG-44, suggesting stronger binding), potential salt bridges (e.g., between ASP-99/ASP-55 and ARG-44), a putative π-π stacking interaction involving TYR-157, and significant hydrophobic interactions from residues such as MET-101 and ALA-112. Collectively, these diverse interactions enable TAT-Nb7 to stably bind to the NS5A antigen with high recognition specificity and strong binding capacity. Detailed information on hydrogen bonds and salt bridges is provided in Appendix A.

## 4. Discussion

BVDV affects more than 80% of cattle worldwide [29]. However, no specific therapeutic drugs are currently available for BVDV infection, and clinical management mainly relies on symptomatic treatment to alleviate symptoms and promote recovery in affected animals [30].Therefore, the development of effective anti-BVDV strategies remains a major global concern.

With advances in genetic engineering, nanobodies have been increasingly applied in various research fields due to their unique structural advantages and thus show great potential in antiviral therapy [31]. Numerous studies have demonstrated that nanobodies exhibit antiviral activity against a range of viruses, including flaviviruses [32], coronaviruses [33], and influenza viruses [34]. Therefore, developing nanobodies as novel antiviral agents against BVDV is of significant importance.

Non-structural proteins are primarily responsible for core viral life processes, including genome replication, transcription, translation regulation, protein processing (e.g., proteases), and immune evasion. These proteins are highly conserved and less prone to drug resistance, making them ideal targets for antiviral therapy. Small-molecule inhibitors targeting HCV NS3 and NS5A have already been approved for clinical use [35,36,37]. BVDV, like HCV, belongs to the Flaviviridae family, and both NS3 and NS5A are relatively conserved and play key roles in regulating BVDV replication [17]. NS3 possesses multiple enzymatic activities, including serine protease, RNA helicase, and nucleoside triphosphatase functions, and is closely involved in viral RNA replication and virus–host interactions [38]. NS5A also plays a central regulatory role in viral RNA replication, assembly, and immune evasion mechanisms [39]. In this study, we expressed and purified highly pure recombinant NS3 and NS5A proteins and used them to immunize alpacas. The immunization induced strong immunogenicity, with high-titer-specific antibodies detected in alpaca serum after the fourth immunization, reaching titers of up to 1:512,000. These results suggest that multi-antigen stimulation may enhance immune cell activation efficiency, and complex cellular interactions can influence the immune response, thereby enhancing the overall immune reactivity [40]. Moreover, immunization with a mixture of multiple antigens may simultaneously activate immune responses against multiple targets, potentially increasing antibody diversity and promoting the generation of antibodies recognizing different epitopes [41].

Phage display technology is one of the most widely used in vitro screening methods for nanobody selection [42]. By inserting nanobody-encoding genes into the reading frame of the phage coat protein gene, the nanobody is fused to the phage coat protein and displayed on the phage surface, enabling efficient selection of antigen-specific clones [43]. This method is not only simple and efficient but also allows direct linkage between genotype and phenotype, making it a key tool for nanobody screening and optimization [44]. We successfully constructed a phage display library with a titer of up to 5.2 × 10^11^ pfu/mL. Studies have shown that libraries with titers ranging from 10^10^ to 10^11^ pfu/mL are considered suitable for efficient screening of high-affinity clones [45], indicating that our library was of ideal scale and quality. Random colony PCR analysis revealed a positive rate of nearly 91%, significantly higher than that of conventional libraries [46]. Sequence analysis of the CDR3 regions of randomly selected clones showed a high degree of sequence repetition, with only a few unique clones identified. Ultimately, four nanobodies targeting BVDV NS5A and three targeting BVDV NS3 were selected. The limited diversity of selected clones may be attributed to the enrichment of high-affinity VHH clones during multiple rounds of biopanning. Clones with stronger binding capacities are preferentially retained, leading to a reduction in overall diversity despite the large library capacity. As a result, the proportion of clones capable of effectively binding to the target antigen remains relatively low [47]. To enhance the diversity of selected nanobodies and improve screening efficiency, future studies may consider optimizing the selection strategy. Potential approaches include incorporating competitive elution [48] or performing multi-round screenings under varied conditions [49], which can provide a more comprehensive assessment of library composition and improve the overall screening outcome.

Binding affinity is a key parameter describing the strength of molecular interactions, used to quantify the tightness of binding between a biomolecule (e.g., an antibody or DNA) and its ligand or receptor (e.g., an antigen or inhibitor) [50]. During the immune response, when B lymphocytes encounter antigens for the first time, the antibodies they produce exhibit low affinity for the antigens. However, with the occurrence of mutations, the affinity of the antibodies gradually increases. This process is referred to as affinity maturation [51]. High-affinity antibodies can reduce effective dosage, enhance targeted delivery, and minimize off-target effects [52]. Therefore, among the selected nanobodies, Nb7 and Nb23, which exhibited the highest affinities, were chosen for further investigation.

However, we observed that the nanobodies had limited intrinsic cell-penetrating ability [53], and MDBK cells suffered from the problem of low transfection efficiency [54]. Thus, effective intracellular delivery of anti-NS3 and anti-NS5A nanobodies remains a critical challenge for their functional application. Cell-penetrating peptides (CPPs) are short peptide sequences capable of crossing cellular membranes while maintaining biological activity [55]. Among them, TAT, derived from the HIV-1 transcriptional activator, is the earliest and most widely used CPP, consisting of an 11-amino-acid peptide that facilitates membrane penetration [56]. TAT interacts with the lipid bilayer and enters cells via direct permeation. Due to its excellent membrane-penetrating ability and low cytotoxicity, TAT has become a promising tool for antiviral drug delivery [57]. In this study, we successfully fused TAT with Nb7 and Nb23, enabling their intracellular delivery, which was confirmed by Western Blot and immunofluorescence assay (IFA).

Further cellular assays demonstrated that TAT-Nb7 significantly inhibited viral replication, whereas TAT-Nb23 exhibited limited inhibitory effects on BVDV replication—despite showing specific binding to the recombinant NS3 protein used in our in vitro experiments. While we cannot rule out the possibility that subtle differences in post-translational modification (e.g., phosphorylation, glycosylation [58]) or conformational states between recombinant NS3 (purified from *E. coli* expression systems) and endogenously expressed NS3 (in BVDV-infected cells) may affect TAT-Nb23’s binding efficiency in vivo, this remains an untested hypothesis. Alternative explanations for its reduced efficacy—such as competition with viral co-factors (e.g., NS4A, which forms a complex with NS3 to enhance its protease activity [11]) or limited access to NS3’s functional domains during viral polyprotein processing—also warrant consideration. To clarify the underlying mechanism, future studies could employ immunofluorescence co-localization or co-immunoprecipitation assays to verify TAT-Nb23’s binding to endogenously expressed NS3 in infected cells, and further explore whether viral co-factors interfere with this interaction.

The novelty of our study lies in targeting BVDV’s NS5A/NS3—proteins indispensable for viral replication—and filling the gap in BVDV-specific therapeutics. Our TAT-Nb7 (targeting NS5A) serves as a promising candidate and provides valuable insights for the development of targeted interventions against BVDV, laying a foundational basis for addressing the unmet critical need for specific antiviral drugs to combat this pathogen.

Although this study confirmed that TAT can effectively deliver nanobodies into cells within a non-toxic concentration range and exert antiviral effects, certain limitations remain. TAT-mediated delivery lacks cell-type specificity and may lead to unintended effects on non-target cells [59]. Identifying novel, tissue-specific delivery systems and validating the antiviral efficacy of the selected nanobodies in animal models are ongoing efforts in our laboratory.

## 5. Conclusions

In conclusion, this study successfully isolated a specific nanobody targeting the BVDV NS5A antigen from an alpaca VHH gene library using phage display technology. A TAT cell-penetrating peptide was fused with the nanobody to generate the TAT-Nb7 protein, which was subsequently expressed and purified. The results demonstrate that TAT-Nb7 is capable of entering cells and effectively inhibiting BVDV replication. These findings provide a potential strategy for the development of novel antiviral agents for the treatment of BVDV infection.

## Figures and Tables

**Figure 1 biomolecules-15-01593-f001:**
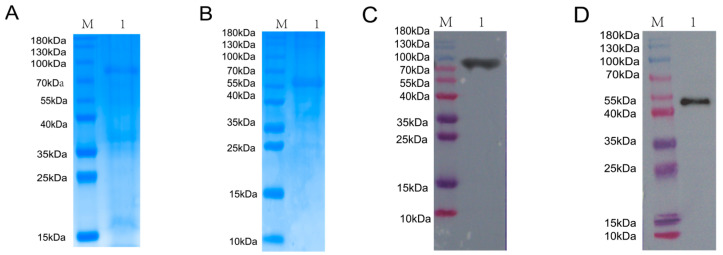
Expression, purification, and identification of BVDV NS3 and NS5A recombinant proteins. (**A**) SDS-PAGE analysis of NS3 recombinant protein expression and purification. M: Protein Marker; Lane 1: purified NS3 protein. (**B**) SDS-PAGE analysis of NS5A recombinant protein expression and purification. M: Protein Marker; Lane 1: purified NS5A protein. (**C**) Western blot verification of purified NS3 protein using an anti-His antibody. M: Protein Marker; Lane 1: purified NS3 protein detected by anti-His antibody. (**D**) Western blot verification of purified NS5A protein using an anti-His antibody.M: Protein Marker; Lane 1: purified NS5A protein detected by anti-His antibody.

**Figure 2 biomolecules-15-01593-f002:**
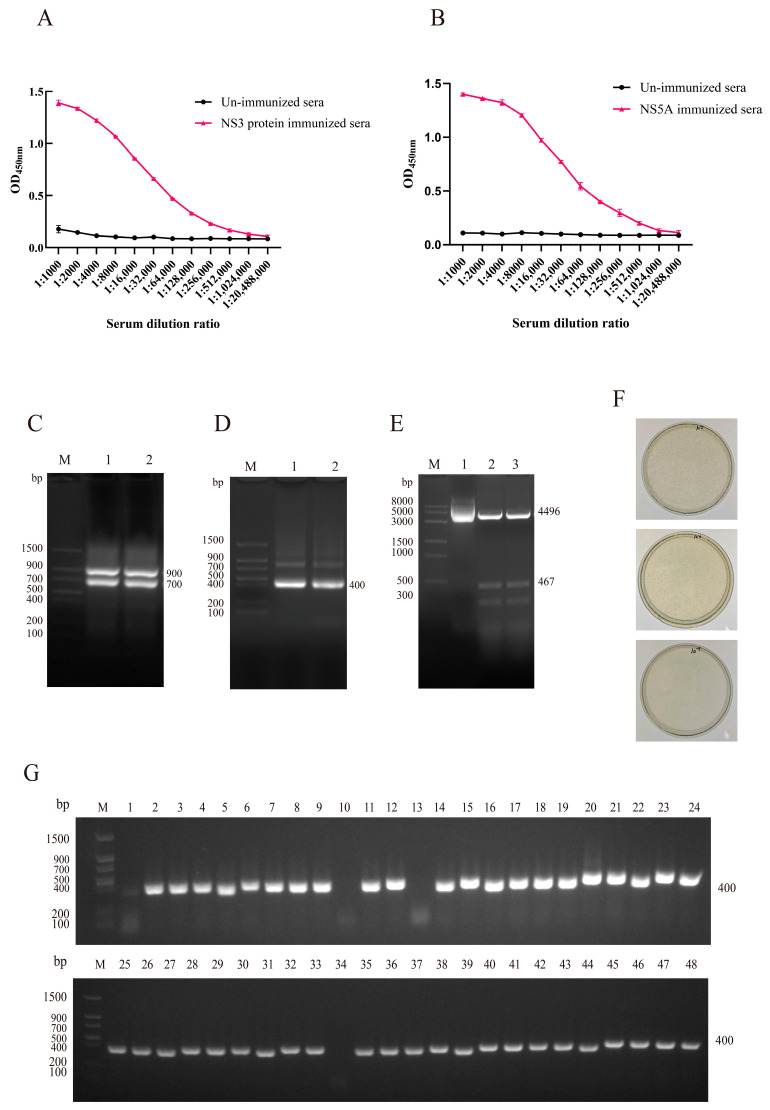
Construction and characterization of the VHH phage display library. (**A**) Antibody titers in post-immunization serum samples were measured by i-ELISA using the NS3 protein. (**B**) Antibody titers in post-immunization serum samples were measured by i-ELISA using the NS5A protein. Data are presented as the mean ± SD of three independent experiments conducted in triplicate. For i-ELISA, technical replicates = 3 (triplicate wells per sample in the ELISA microplate). (**C**) First-round PCR amplification yielded a 700 bp DNA fragment. M: Protein Marker; Lane 1 and Lane 2: First-round PCR products (700 bp). (**D**) Second-round PCR produced a VHH-specific fragment. M: Protein Marker; Lane 1 and Lane 2: Second-round PCR products (400 bp). (**E**) Double-enzyme digestion confirmed successful ligation of the VHH fragment into the pCANTAB-5E vector. M: DNA Marker; Lane 1: Undigested recombinant plasmid; Lane 2–3: After digestion with restriction enzymes, two bands were generated: the vector fragment (4496 bp) and the inserted target DNA fragment (467 bp). (**F**) Library titer was calculated by plate counting and determined to be 5.2 × 10^11^ pfu/mL. (**G**) PCR screening of 48 randomly selected clones showed an insertion efficiency of 91% (44 out of 48 clones).

**Figure 3 biomolecules-15-01593-f003:**
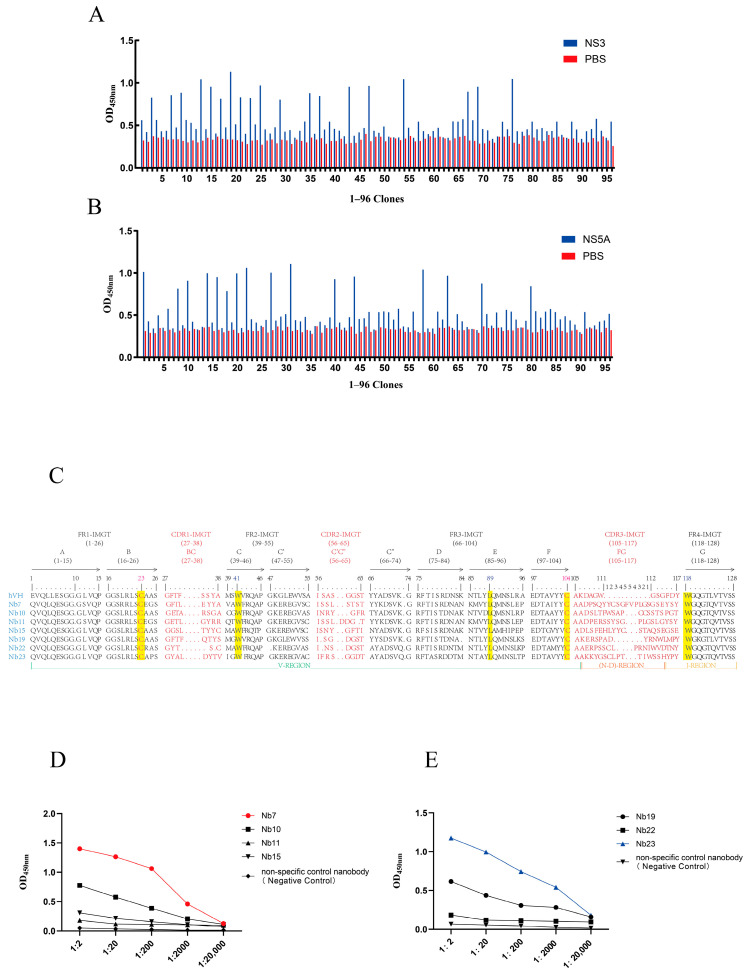
Specificity and reactivity of isolated nanobodies (Nbs) against NS3 and NS5A proteins. (**A**) i-ELISA analysis of periplasmic extracts from 96 clones for reactivity against the NS3 protein. (**B**) i-ELISA analysis of periplasmic extracts from 96 clones for reactivity against the NS5A protein. (**C**) Amino acid sequence alignment of CDR3 regions from isolated nanobodies with human VH sequences, using the IMGT unique numbering system (PMID: 12477501) and IMGT-delimited framework (FR-IMGT) and complementarity determining (CDR-IMGT) regions [28]. (**D**) i-ELISA assessing the binding affinity of selected nanobodies to the NS5A protein. A non-specific control nanobody targeting PRRSV served as a negative control. (**E**) i-ELISA assessing the binding affinity of selected nanobodies to the NS3 protein. A non-specific control nanobody targeting PRRSV served as a negative control. Data are presented as the mean ± SD of three independent experiments conducted in triplicate.

**Figure 4 biomolecules-15-01593-f004:**
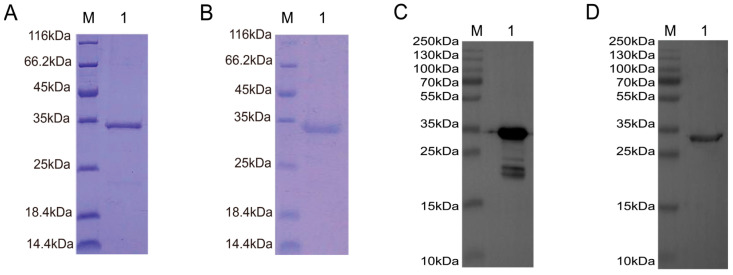
Expression, purification, and identification of TAT-Nb7 and TAT-Nb23 nanobodies. (**A**) SDS-PAGE analysis of purified TAT-Nb7. M: Protein Marker; Lane 1: Purified TAT-Nb7. (**B**) SDS-PAGE analysis of purified TAT-Nb23. M: Protein Marker; Lane 1: Purified TAT-Nb23. (**C**) Western Blot analysis of purified TAT-Nb7 using an anti-His antibody. M: Protein Marker; Lane 1: Purified TAT-Nb7 detected by anti-His antibody. (**D**) Western Blot analysis of purified TAT-Nb23 using an anti-His antibody. M: Protein Marker; Lane 1: Purified TAT-Nb23 detected by anti-His antibody.

**Figure 5 biomolecules-15-01593-f005:**
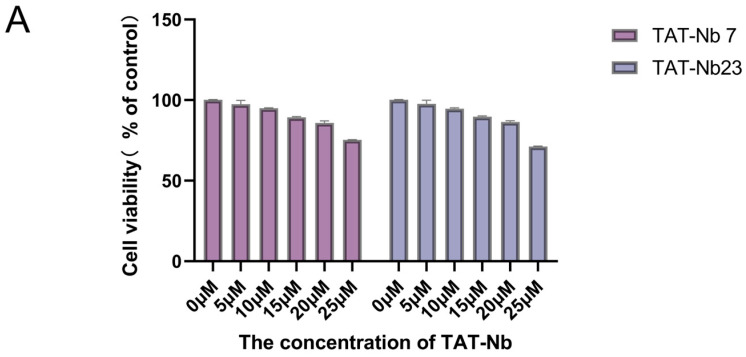
Cytotoxicity analysis and transmembrane verification of TAT-Nbs. (**A**) Effect of TAT-Nb7 and TAT-Nb23 proteins on the viability of MDBK cells. Data are presented as the mean ± SD of three independent experiments conducted in triplicate. For this experiment, biological replicates (independent cell culture batches) = 3, and technical replicates (duplicate wells per sample in 96-well plates) = 8. (**B**) Verification of the transmembrane efficiency of TAT-Nb7 and TAT-Nb23 via Western Blotting. (**C**) Verification of the transmembrane efficiency of TAT-Nb7 and TAT-Nb23 via IFA. Magnification, ×100; scale bars, 200 μm.

**Figure 6 biomolecules-15-01593-f006:**
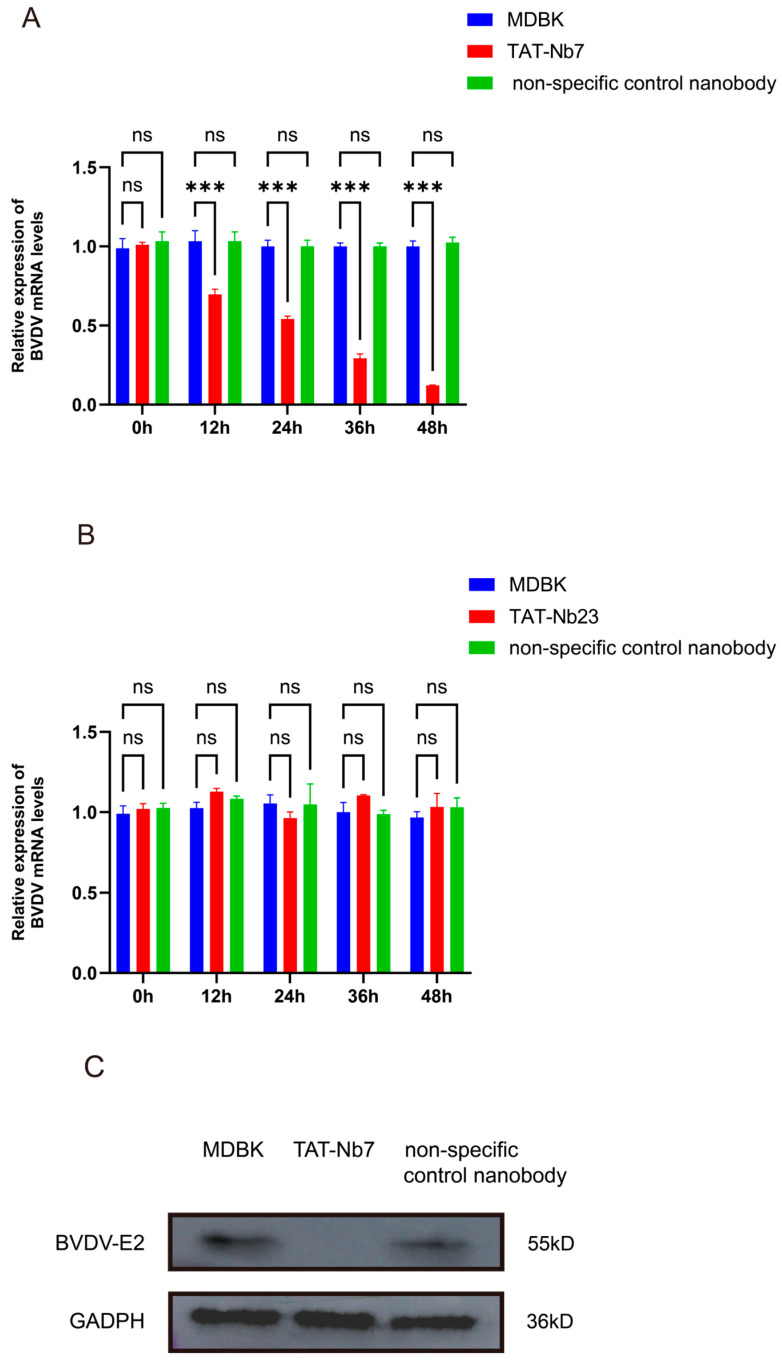
The effect of TAT Nbs on BVDV replication in MDBK cells. (**A**) Detection of the effect of TAT-Nb7 on BVDV replication through RT-qPCR. (**B**) Detection of the effect of TAT-Nb23 on BVDV replication through RT-qPCR. MDBK: Blank control group infected with BVDV only without therapeutic treatment; TAT-Nb7: Experimental group infected with BVDV followed by treatment with TAT-Nb7; TAT-Nb23: Experimental group infected with BVDV followed by treatment with TAT-Nb23; Non specific control nanobody: using unrelated nanoparticles targeting PRRSV as a control. Data were calculated using the 2^−ΔΔCT^ method and analyzed by one-way ANOVA; *** *p* < 0.001; ns, not significant. Error bars represent the standard deviation (SD) of three independent experiments. Experiments involved 3 biological replicates (each representing a separate BVDV infection event in MDBK cells) and 3 technical replicates per biological sample (triplicate qPCR reactions for each RNA sample). (**C**) Detection of the effect of TAT-Nb7 on BVDV replication at 48 h post viral challenge using Western Blot. MDBK: Blank control group infected with BVDV only without therapeutic treatment; TAT-Nb7: Experimental group infected with BVDV followed by treatment with TAT-Nb7; Non specific control nanobody: using unrelated nanoparticles targeting PRRSV as a control.

**Figure 7 biomolecules-15-01593-f007:**
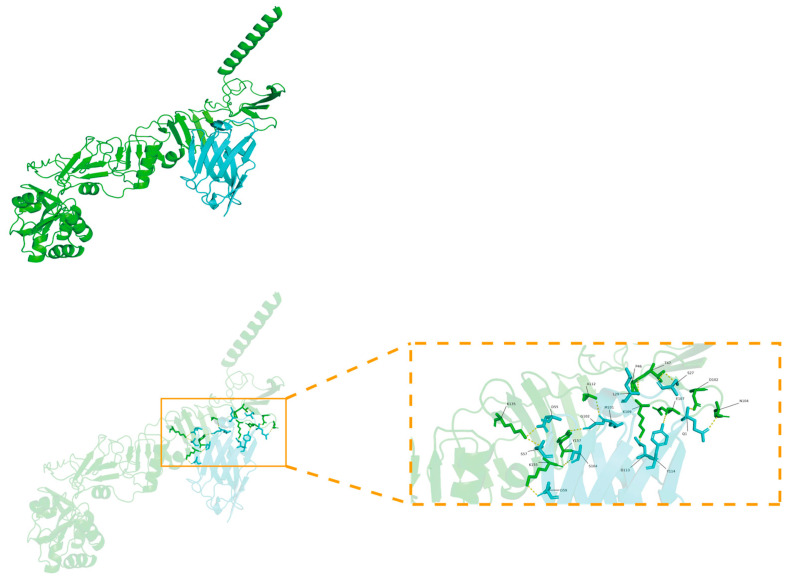
Molecular docking model of TAT-Nb7 and NS5A. The NS5A antigen protein appears green, while TAT-Nb7 appears blue. Key residues are labeled and represented as rod-shaped models, with hydrogen bonds indicated by dashed lines.

**Table 1 biomolecules-15-01593-t001:** Primers required for gene amplification of object.

Primers	Primer Sequences (5′–3′)	Digestion Site
AIPVh-LD	CTTGGTGGTCCTGGCTGC	
CH2-R	GGTACGTGCTGTTGAACTGTTCC	
VHH-Forward	TCGCGGCCCAGCCGGCCCAGGTCCAACTGCAGGAGTCTGGGG	*Sfi I*
VHH-Reverse	ATAAGAATGCGGCCGCTGAGGAGACGGTGACCTGGGTCCCC	*Not I*

**Table 2 biomolecules-15-01593-t002:** Primers for RT-qPCR.

Primers	Primer Sequences (5′–3′)
β-actin F	TGCTGTCCCTGTATGCCTCT
β-actin R	TGTCACGCACGATTTCCC
5′UTR-F	CCTAGCCATGCCCTTAGTAGGACT
5′UTR-R	GGAACTCCATGTGCCATGTACA

F: Positive primer. R: Reverse primer.

**Table 3 biomolecules-15-01593-t003:** Enrichment of specific phage during NS3 protein screening.

Elution Rounds	Phage Input (PFU/mL)	P Output (PFU/mL)	N Output (PFU/mL)	Recovery (P/Input)	P/N
First Round	3.5 × 10^12^	2.4 × 10^4^	3.6 × 10^3^	6.86 × 10^−9^	6.7
Second Round	3.5 × 10^12^	7.9 × 10^5^	2.7 × 10^4^	2.3 × 10^−7^	29.2
Third Round	3.5 × 10^12^	3.01 × 10^6^	3.6 × 10^4^	8.6 × 10^−7^	83.6

P: NS3 protein-coated well. N: control (blocked-only) well. P/N: The ratio of phage output from the antigen-coated well to the control well.

**Table 4 biomolecules-15-01593-t004:** Enrichment of specific phage during NS5A protein screening.

Elution Rounds	Phage Input (PFU/mL)	P Output (PFU/mL)	N Output (PFU/mL)	Recovery (P/Input)	P/N
First Round	4.5 × 10^12^	1.88 × 10^5^	5.5 × 10^4^	4.18 × 10^−8^	3.4
Second Round	4.5 × 10^12^	6.4 × 10^6^	9.1 × 10^4^	1.42 × 10^−6^	70.3
Third Round	4.5 × 10^12^	8.5 × 10^7^	7.5 × 10^5^	1.89 × 10^−5^	113.3

P: NS5A protein-coated well. N: control (blocked-only) well. P/N: The ratio of phage output from the antigen-coated well to the control well.

## Data Availability

The original contributions presented in this study are included in the article/Appendix A. Further inquiries can be directed to the corresponding author.

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
