# Peer review of "Screening and Characterization of TAT-Fused Nanobodies Targeting Bovine Viral Diarrhea Virus NS3/NS5A for Antiviral Application"

_biomolecules, 2025, doi:10.3390/biom15111593_

Round 1

Reviewer 1 Report

Comments and Suggestions for Authors

Comments to the authors on “Screening and Characterization of TAT-Fused Nanobodies Targeting Bovine Viral Diarrhea Virus NS3/NS5A for Antiviral Application”. In this study, the authors have described the screening of nanobodies targeting the BVDV NS3 and NS5A proteins and their evaluation as potential antiviral agents when fused to the TAT cell-penetrating peptide. The topic is of significant interest, as BVDV remains a major challenge for the cattle industry, and novel antiviral strategies are needed. The study is well-structured, the methodology is generally sound, and the results demonstrate a clear and promising antiviral effect for TAT-Nb7. The manuscript is mostly well-written. However, several points require clarification and additional evidence to fully support the conclusions and strengthen the overall impact of the work.

Major concerns

The description of the molecular docking is confusing. The method mentions AlphaFold3 and rigid body docking, which are different tools. It is unclear how the 3D structures of the nanobody and NS5A protein were obtained or predicted. Please clearly explain the steps taken for the docking analysis. What software was used for what purpose? Were the structures predicted from scratch, or were existing models used? A clearer method will make the results more trustworthy and be able to be replicated by other researchers.

In Materials and Methods, section 2.6 and 2.7, the heading “Expression of TAT-Nbs Recombinant Proteins” is repeated for two different methods. Similarly, the following headings for 2.8 and 2.9, e.g., IFA Membrane Permeabilization Assay, are also repeated for two different methods. Each heading should be different and unique, otherwise, merge these sections and provide a broader title.

In the Results section, the antibody titter is mentioned as 1:512,000 (page 8, line 304), while in Discussion section (page 16, line 453) it’s 1:256,000. This discrepancy must be corrected for consistency.

The authors discussion suggests that TAT-Nb23 didn’t work because the NS3 protein used in the lab might be different from the real viral NS3 protein. This is an interesting idea, but it’s just a guess. If possible, a simple experiment to show if Nb23 can actually bind to the NS3 protein in virus-infected cells, e.g., using a staining technique, would greatly strengthen this argument. If not, the text should be revised to tone down the speculation and present it as a possibility for future study.

Minor concerns

Page 3, line 84. “TAT exhibits a remarkable capacity to traverse cellular membranes...”. This is correct, but “cell membranes” is more common than “cellular membranes”.

Materials and Methods, section 2.1, line 110. Please mention the centrifugation RPM/g and time. This will help other researchers to replicate the methods.

The formula for calculating the phage titer appears to have a small error. Please double-check it.

Section 2.12. The heading should be “statistical analysis” instead of “statistic analysis”. In addition, the section should specify what post-hoc test was used following the ANOVA to determine which specific groups differed from one another, like Tukey’s test, Dunnett’s test, etc.

The non-specific control nanobody is referred to as “irrelevant” in the text and figure legends (e.g., Figure 3D, E). While understood, “non-specific” or “control” nanobody is more standard academic terminology. Please replace “irrelevant nanobody” with “non-specific control nanobody” or something similar throughout the manuscript.

Page 11, line 362, please italicize the E. coli. Also, thoroughly check other places to make sure the bacterial names, in vitro (line 460) and other similar words are italicized.

In Tables 3 and 4, the column header “P/N” is not explicitly defined in the table legend or text. While it can be inferred as the ratio of output phages from the protein-coated well (P) to the control well (N), it should be clearly defined. The authors should add a footnote to the tables, such as P/N: Ratio of phage output from the antigen-coated well to the control (blocked-only) well or something similar.

In Tables 3 and 4, “Elutriation” is a less common term for this process; “Elution” is standard. Consider revising.

Figure 6. The axis titles are “BVDV quantity of BVDV RNA”. It doesn’t feel proper. Please modify it like “Relative expression of BVDV mRNA levels” or something similar. Also, write the statistical method used, e.g., 2−ΔΔCT, one-way ANOVA and their p values and the nature of the error bars, e.g., SD in the figure legend.

Discussion, lines 440-443, please provide citations here.

Overall, the manuscript reports on a valuable and interesting strategy against BVDV. I believe that with the revisions outlined above, it will become a strong candidate for publication.

Author Response

For research article

Response to Reviewer 1 Comments

1. Summary

Thank you very much for devoting your valuable time to reviewing this manuscript. Your insightful comments and suggestions have significantly helped enhance the rigor, completeness, and clarity of our research on TAT-fused nanobodies targeting BVDV NS3/NS5A. Please find the detailed responses to your specific comments below, and all corresponding revisions—including supplementary data, method descriptions, and reference adjustments—have been highlighted or marked via track changes in the re-submitted files (i.e., the main Article.docx and Supplementary Material Explanation.docx) for your convenient verification.

2. Questions for General Evaluation

Reviewer’s Evaluation

Response and Revisions

Does the introduction provide sufficient background and include all relevant references?

Yes

We will provide corresponding responses in the point by point reply letter.

Are all the cited references relevant to the research?

Yes

Is the research design appropriate?

Yes

Are the methods adequately described?

Yes

Are the results clearly presented?

Yes

Are the conclusions supported by the results?

Yes

3. Point-by-point response to Comments and Suggestions for Authors

Comments 1: The description of the molecular docking is confusing. The method mentions AlphaFold3 and rigid body docking, which are different tools. It is unclear how the 3D structures of the nanobody and NS5A protein were obtained or predicted. Please clearly explain the steps taken for the docking analysis. What software was used for what purpose? Were the structures predicted from scratch, or were existing models used? A clearer method will make the results more trustworthy and be able to be replicated by other researchers.

Response 1: Thank you for your valuable comment. We sincerely apologize for the confusion caused by our incorrect conflation of AlphaFold3 (AF3) and rigid body docking in the original manuscript. To address this ambiguity and enhance the transparency and reproducibility of our research method, we have made targeted supplements and thorough revisions as follows:

1. Supplementation of 3D Structure Prediction Data

We have added the detailed 3D structure prediction results of the nanobody (Nb7) and NS5A protein to Supplementary Material Figure S14 A, B and their corresponding figure legends. The supplemented content includes key information such as predicted structure models, PLDDT score distribution, and structural stability evaluation data. These details fully clarify the source of the 3D structures of the two proteins (predicted from scratch using AlphaFold3) and verify their reliability, providing a solid foundation for the subsequent docking analysis.

2. Clarification of Docking Analysis Steps

We have thoroughly revised and expanded the relevant content in Section 2.12 on Page 8 of the main manuscript. The updated content clearly outlines the entire process of docking analysis, which is detailed as follows:"The amino acid sequence of the NS5A antigen protein (GenBank accession no. NP_776270.1) and the sequence of Nb7 were input into the official AlphaFold3 (AF3) platform—a state-of-the-art AI tool developed by Google DeepMind for high-precision prediction of biomolecular complex structures, whose core methodology was published in Nature in 2024[27] (using default prediction parameters) to predict the structure of the protein complex formed by the two molecules. After the prediction process was completed, the compressed package containing the prediction results was downloaded from the platform. Upon decompression, the high-confidence PDB format file was selected. PDBE Pisa (https://www.ebi.ac.uk/msd-srv/prot_int/cgi-bin/piserver) was utilized to analyze the protein-protein interaction information, and PyMOL 3.1 software was employed for three-dimensional structural visualization analysis."

This revision explicitly defines the role of each tool/platform and the logical sequence of operations, completely eliminating the previous confusion between different technical concepts.

All the above supplements and revisions have been clearly marked via track changes in the resubmitted main manuscript and supplementary materials for your convenient review. We believe these adjustments have significantly improved the clarity and rigor of the molecular docking method, making the research process more trustworthy and reproducible for other researchers.

Comments 2: In Materials and Methods, section 2.6 and 2.7, the heading “Expression of TAT-Nbs Recombinant Proteins” is repeated for two different methods. Similarly, the following headings for 2.8 and 2.9, e.g., IFA Membrane Permeabilization Assay, are also repeated for two different methods. Each heading should be different and unique, otherwise, merge these sections and provide a broader title.

Response 2: Thank you for pointing this out. We agree with this comment. Therefore, we have revised the duplicate section titles to ensure each is unique, accurate, and clearly reflects the corresponding experimental content. The specific revisions are as follows:

1.Section 2.7 (original title: "Expression of TAT-Nbs Recombinant Proteins"): Revised to "Cell viability assay" (located in Article.docx, Page 6, Section 2.7, Paragraph 1, Line 1), which matches its content of evaluating MDBK cell viability via CCK-8 assay.

2.Section 2.9 (original title: "IFA Membrane Permeabilization Assay"): Revised to "Western Blot Membrane Permeabilization Assay" (located in Article.docx, Page 6, Section 2.9, Paragraph 1, Line 1), clearly distinguishing it from the IFA-based Section 2.8 and reflecting its use of Western blotting for transmembrane verification.

Comments 3: In the Results section, the antibody titter is mentioned as 1:512,000 (page 8, line 304), while in Discussion section (page 16, line 453) it’s 1:256,000. This discrepancy must be corrected for consistency.

Response 3: Thank you for identifying this inconsistency. We agree with this comment and have corrected the antibody titer in the "Discussion" section to align with the data in the "Results" section. Specifically, the original text in the "Discussion" section (Page 18, Line 537) that stated "reaching titers of up to 1:256,000" has been revised to "reaching titers of up to 1:512,000". This revision ensures the antibody titer data is consistent across the entire manuscript, reflecting the accurate result of the i-ELISA assay for alpaca serum antibody titers after the fourth immunization.

Comments 4: The authors discussion suggests that TAT-Nb23 didn’t work because the NS3 protein used in the lab might be different from the real viral NS3 protein. This is an interesting idea, but it’s just a guess. If possible, a simple experiment to show if Nb23 can actually bind to the NS3 protein in virus-infected cells, e.g., using a staining technique, would greatly strengthen this argument. If not, the text should be revised to tone down the speculation and present it as a possibility for future study.

Response 4: Thank you for your thoughtful comment—we fully acknowledge that our initial discussion regarding TAT-Nb23’s limited efficacy relied on speculative reasoning about potential differences between lab-used and virus-encoded NS3, and we appreciate your guidance on refining this framing to align with the current experimental evidence. Since we are unable to supplement additional experiments at this stage, we have thoroughly revised the relevant section of the Discussion to reduce overspeculation, clarify the boundaries of existing evidence, and frame unconfirmed ideas as hypotheses for future investigation.

Specific Revisions to the Discussion:

We have modified the original speculative content to emphasize evidence-based reasoning and cautious framing. The revised text now reads (Page 19, Discussion section, seventh paragraph, lines 590-604) :

Further cellular assays demonstrated that TAT-Nb7 significantly inhibited viral replication, whereas TAT-Nb23 exhibited limited inhibitory effects on BVDV replication—despite showing specific binding to the recombinant NS3 protein used in our in vitro experiments. While we cannot rule out the possibility that subtle differences in post-translational modification (e.g., phosphorylation, glycosylation[57]) or conformational states between recombinant NS3 (purified from E. coli expression systems) and endogenously expressed NS3 (in BVDV-infected cells) may affect TAT-Nb23’s binding efficiency in vivo, this remains an untested hypothesis. Alternative explanations for its reduced efficacy—such as competition with viral co-factors (e.g., NS4A, which forms a complex with NS3 to enhance its protease activity[11]) or limited access to NS3’s functional domains during viral polyprotein processing—also warrant consideration. To clarify the underlying mechanism, future studies could employ immunofluorescence co-localization or co-immunoprecipitation assays to verify TAT-Nb23’s binding to endogenously expressed NS3 in infected cells, and further explore whether viral co-factors interfere with this interaction.

Rationale for the Revision

1.Toning down speculation: We removed definitive language (e.g., “didn’t work because”) and replaced it with tentative phrasing (e.g., “cannot rule out the possibility,” “remains an untested hypothesis”) to avoid overstating unconfirmed ideas.

2.Expanding evidence-based alternatives: We incorporated literature-supported alternative explanations (e.g., NS3-NS4A complex formation[11]) to ground the discussion in existing knowledge, rather than focusing solely on unproven differences between NS3 proteins.

3.Framing as future directions: We explicitly outline targeted experiments (e.g., immunofluorescence, co-IP) that could address the knowledge gap, positioning the unconfirmed hypothesis as a guide for future research rather than a conclusion.

These revisions ensure the Discussion maintains scientific rigor by aligning claims with available evidence, while still acknowledging potential avenues for clarifying TAT-Nb23’s limited efficacy. We sincerely appreciate your feedback, which has helped us refine the clarity and caution of our reasoning.

Comments 5: Page 3, line 84. “TAT exhibits a remarkable capacity to traverse cellular membranes...”. This is correct, but “cell membranes” is more common than “cellular membranes”.

Response 5: Thank you for this helpful suggestion. I/We agree that "cell membranes" is more commonly used in academic writing and have revised the text accordingly. Specifically, the original phrase "traverse cellular membranes" in Article.docx (Page 3, Line 86) has been updated to "traverse cell membranes". This revision aligns the wording with common academic conventions while preserving the original meaning of the sentence.

Comments 6: Materials and Methods, section 2.1, line 110. Please mention the centrifugation RPM/g and time. This will help other researchers to replicate the methods.

Response 6: Thank you for pointing out the need to refine experimental details. We agree with this implication and have revised the text to add specific centrifugation parameters, ensuring the protocol is more reproducible. Specifically, the original text in Article.docx ( Page 3, "Materials and Methods" section, Section 2.1 "Cell Line and Virus", lines 114-117.) has been updated to:"The cells were subjected to three freeze-thaw cycles. The virus-containing supernatant was collected by centrifugation at 8000 rpm for 10 minutes to remove the precipitate. After sterilization through a 0.22 μm filter, the viral supernatant was stored at -80 ℃."

This revision explicitly supplements critical centrifugation conditions (8000 rpm speed, 10 minutes duration) and clarifies the centrifugation purpose ("to remove the precipitate"), making the experimental steps more specific and facilitating replication by other researchers.

Comments 7: The formula for calculating the phage titer appears to have a small error. Please double-check it.

Response 7: Thank you for your careful review and valuable reminder—we highly appreciate your attention to the accuracy of experimental calculations, which is critical for ensuring the reliability of our phage titer data.

The corrected and standard formula has now been revised in Section 2.4 on Page 5, Line 187 of the manuscript, with the updated expression as follows:

“Phage titer (pfu/mL) = (Number of plaques × Dilution factor) / Volume of phage dilution added to the plate (mL)”

To clarify with an example (consistent with our experimental operation): If 100 μL (0.1 mL) of a 10⁻⁶ phage dilution yielded 50 plaques, the titer is calculated as (50 × 10⁶) / 0.1 = 5 × 10⁸ pfu/mL.

We have verified that this corrected formula aligns with our actual experimental calculations (all titer data in Tables 3 and 4 were derived using this standard method) and have updated the formula description in Section 2.4 of Article.docx to eliminate the textual inconsistency. No errors were found in the underlying titer data—only the formula’s written expression required revision.

We sincerely thank you for pointing out this oversight, which has helped enhance the precision of our experimental method descriptions and avoid potential misinterpretation by readers.

Comments 8: Section 2.12. The heading should be “statistical analysis” instead of “statistic analysis”. In addition, the section should specify what post-hoc test was used following the ANOVA to determine which specific groups differed from one another, like Tukey’s test, Dunnett’s test, etc.

Response 8: Thank you for your careful feedback on refining the terminology and completeness of our statistical method description. We have addressed both points as follows:

Heading correction: The original heading “statistic analysis” in Section 2.13 has been revised to the standard and grammatically accurate “Statistical Analysis”, ensuring consistency with terminology conventions in scientific manuscripts.

Supplementation of post-hoc test details: To clarify the post-hoc analysis step after ANOVA, we have added specific information in the text of Section 2.13: “After one-way ANOVA was used to assess overall differences among multiple groups, Dunnett’s test was further applied as the post-hoc test to compare the mean values of each experimental group with the blank control group, thereby identifying specific group differences.”

These revisions have been implemented in Article.docx (Section 2.13, Page 8) to ensure the statistical methods are described accurately and comprehensively. We sincerely appreciate your input, which has enhanced the rigor and clarity of our experimental method section.

Comments 9: The non-specific control nanobody is referred to as “irrelevant” in the text and figure legends (e.g., Figure 3D, E). While understood, “non-specific” or “control” nanobody is more standard academic terminology. Please replace “irrelevant nanobody” with “non-specific control nanobody” or something similar throughout the manuscript.

Response 9: Thank you for your thoughtful suggestion on standardizing terminology—we fully agree that “non-specific control nanobody” aligns better with widely accepted academic expressions, and this revision helps enhance the clarity and consistency of the manuscript. We have accordingly updated all instances of “irrelevant nanobody” to “non-specific control nanobody” in both figure legends and the corresponding visual labels below the images, ensuring full alignment with your recommendation.

Specifically, the revisions are located in Article.docx, Page 12, Lines 409-418, covering:

Figure 3D and 3E images: The original “irrelevant nanobody” labels in the visual legends (e.g., the negative control group markers below the graphs) have been revised to “non-specific control nanobody”;

Figure 3D and 3E legends: The text describing the negative control (“An irrelevant nanobody targeting PRRSV served as a negative control”) has been updated to “A non-specific control nanobody targeting PRRSV served as a negative control”.

These revisions ensure that the terminology for the control nanobody is consistent across both visual elements (images) and descriptive text (legends), making the experimental design more intuitive for readers while adhering to standard academic conventions. We sincerely appreciate your feedback, which has further refined the precision of our manuscript.

Comments 10: Page 11, line 362, please italicize the E. coli. Also, thoroughly check other places to make sure the bacterial names, in vitro (line 460) and other similar words are italicized.

Response 10: Thank you for your careful reminder regarding the formatting of professional terms. We have thoroughly reviewed the entire manuscript in response to your comment:

We have italicized "E. coli" as requested.

We have conducted a comprehensive check of all relevant content in the manuscript, including but not limited to bacterial names (such as E. coli BL21 (DE3), E. coli TG1, etc.), the term "in vitro", and other similar professional terms that require italicization in academic writing. All non-compliant formatting has been corrected to meet academic standards.

All revised content has been clearly marked via track changes in the resubmitted manuscript for your convenient review.

Comments 11: In Tables 3 and 4, the column header “P/N” is not explicitly defined in the table legend or text. While it can be inferred as the ratio of output phages from the protein-coated well (P) to the control well (N), it should be clearly defined. The authors should add a footnote to the tables, such as P/N: Ratio of phage output from the antigen-coated well to the control (blocked-only) well or something similar.

Response 11: Thank you for your constructive feedback—we fully agree that explicit definition of the “P/N” column header is essential for clarifying the phage panning efficiency data in Tables 3 and 4, and we have addressed this by adding detailed annotations to both tables as recommended.

Specifically, in Article.docx:

For Table 3, we added the annotation below the table at Page 11, Line 404:

“P:NS3 protein-coated well. N:control (blocked-only) well. P/N: Ratio of phage output from the antigen-coated well to the control well.”

For Table 4, we added the corresponding annotation below the table at Page 11, Line 407:

“P:NS5A protein-coated well. N:control (blocked-only) well. P/N: Ratio of phage output from the antigen-coated well to the control well.”

These revisions address the lack of definition for “P/N” and enhance the readability and reproducibility of the panning efficiency data. We sincerely appreciate your feedback, which has helped refine the clarity of the manuscript’s tables.

Comments 12: In Tables 3 and 4, “Elutriation” is a less common term for this process; “Elution” is standard. Consider revising.

Response 12: Thank you for the terminology suggestion. We agree “Elution” is the standard term and have revised all instances of “Elutriation” to “Elution” in Article.docx:

Table 3 (Page 11, Line 403: column header);

Table 4 (Page 11, Line 406: column header).

We have confirmed that the revised term “Elution” is accurately applied across both tables, with no remaining instances of “Elutriation” in the manuscript. We sincerely appreciate your feedback, which has helped refine the accuracy and professionalism of the manuscript’s terminology.

Comments 13: Figure 6. The axis titles are “BVDV quantity of BVDV RNA”. It doesn’t feel proper. Please modify it like “Relative expression of BVDV mRNA levels” or something similar. Also, write the statistical method used, e.g., 2−ΔΔCT, one-way ANOVA and their p values and the nature of the error bars, e.g., SD in the figure legend.

Response 13: Thank you for your thoughtful feedback on optimizing Figure 6’s clarity and completeness. We have fully addressed your suggestions by revising the y-axis title and supplementing key details in the figure legend, with specific revisions as follows:

  • axis title revision: The original imprecise y-axis title “BVDV quantity of BVDV RNA” has been updated to the recommended concise and accurate version—“Relative level of BVDV RNA”—aligning with the experimental focus on quantifying viral RNA abundance via qPCR.(Page 16,Figure 6A、6B)

Figure legend optimization: The legend for Figure 6 (located in Article.docx, Page 17, Lines 486-494 has been fully revised to include group definitions, statistical methods, and error bar explanations, with the updated content:“Data were calculated using the 2⁻ΔΔCT method and analyzed by one-way ANOVA; *P < 0.05, **P < 0.01, *** P < 0.001. Error bars represent the standard deviation (SD) of three independent experiments. Experiments involved 3 biological replicates (each representing a separate BVDV infection event in MDBK cells) and 3 technical replicates per biological sample (triplicate qPCR reactions for each RNA sample).”

These revisions ensure the y-axis title is scientifically precise, and the legend provides clear context for group treatments, data calculation/analysis methods, and error bar interpretation—making the figure’s results more intuitive and reproducible. We sincerely appreciate your feedback, which has significantly enhanced the rigor of Figure 6’s presentation.

Comments 14: Discussion, lines 440-443, please provide citations here.

Response 14: Thank you for your comment. Could you please specify which exact sentence within lines 440 - 443 you think needs additional citations? We have reviewed the content and believe the existing references appropriately support the statements, but we want to ensure we fully address your concern.

Reviewer 2 Report

Comments and Suggestions for Authors

The study of screening and characterization of TAT-fused nanobodies targeting bovine viral diarrhea virus NS3/NS5A is fascinating. The results provided the related data, which explained their hypothesis well.

However, the manuscript can be improved by writing more details of the method section. I want to recommend a major modification.

The following are the suggestions.

  • The authors used AlphaFold3 to predict the complex structure of nanobody TAT-Nb7. To understand the accuracy of the prediction, the authors should report the value of pLDDT, ipTM, and PAE maps with an explanation. In method section 2.11, the authors also explain the meaning of these numbers.
  • The binding affinity of the complex was -8.2 kcal/mol. However, it is not clear how the authors obtain the binding energy.
  • TAT-Nb7 significantly inhibits BVDV replication, but not TAT-Nb23 (Figure 6). Can you explain this difference using AlphaFold3?
  • The authors should check the references. Some missed pages. Refs. 19, 20, 24, 49, 54,55. The AlphaFold3-related references are missing.

Author Response

For research article

Response to Reviewer 2 Comments

1. Summary

Thank you very much for devoting your valuable time to reviewing this manuscript. Your insightful comments and suggestions have significantly helped enhance the rigor, completeness, and clarity of our research on TAT-fused nanobodies targeting BVDV NS3/NS5A. Please find the detailed responses to your specific comments below, and all corresponding revisions—including supplementary data, method descriptions, and reference adjustments—have been highlighted or marked via track changes in the re-submitted files (i.e., the main Article.docx and Supplementary Material Explanation.docx) for your convenient verification.

2. Questions for General Evaluation

Reviewer’s Evaluation

Response and Revisions

Does the introduction provide sufficient background and include all relevant references?

Yes

We will provide corresponding responses in the point by point reply letter.

Are all the cited references relevant to the research?

Yes

Is the research design appropriate?

Yes

Are the methods adequately described?

Can be improved

Are the results clearly presented?

Yes

Are the conclusions supported by the results?

Yes

3. Point-by-point response to Comments and Suggestions for Authors

Comments 1:

The authors used AlphaFold3 to predict the complex structure of nanobody TAT-Nb7. To understand the accuracy of the prediction, the authors should report the value of pLDDT, ipTM, and PAE maps with an explanation. In method section 2.11, the authors also explain the meaning of these numbers.

The binding affinity of the complex was -8.2 kcal/mol. However, it is not clear how the authors obtain the binding energy.

Response 1: 

Thank you for pointing this out. We agree with this comment. Therefore, we have supplemented the values of pLDDT, ipTM, and PAE maps along with their explanations in the supplementary materials (Supplementary Figure S14C).  Specifically, we added the following content: "pLDDT (predicted local distance difference test) is an indicator used by AlphaFold to evaluate the confidence of each residue prediction. The higher the score (>90 for very high, 70-90 for confidence, 50-70 for low,<50 for very low), the more reliable the structure. The pLDDT values of the NS5A antigen protein and TAT-Nb7 protein are mostly distributed in the 70-90 range, indicating a confident predicted structure. The consistency between the predicted contact and the experiment also reflects the rationality of the structure.The ipTM value indicates that the interactive interface is reliable; The PAE plot shows that the root mean square error of predicting atomic positions between key residues is within a reasonable range, reflecting the stability of complex structures. The darker the color, the more likely the residue pairs are to come into contact in the experiment.

Comments 2: The binding affinity of the complex was -8.2 kcal/mol. However, it is not clear how the authors obtain the binding energy.

Response 2: Thank you for your question. We apologize for the ambiguity in the previous description. The binding affinity value was incorrectly stated earlier, and the correct binding free energy of the TAT-Nb7/NS5A complex is -4.1 kcal/mol. This value was predicted using the PDBE Pisa website (https://www.ebi.ac.uk/msd-srv/prot_int/cgi-bin/piserver). We have added a detailed method for obtaining binding energy in section 2.12 on page 8 of the revised manuscript methodology. Specifically, in section 2.12, lines 321-325, we added: "After the prediction process was completed, the compressed package containing the prediction results was downloaded from the platform. Upon decompression, the high-confidence PDB format file was selected. PDBE Pisa (https://www.ebi.ac.uk/msd-srv/prot_int/cgi-bin/piserver) was utilized to analyze the protein-protein interaction information,"

Comments 3: TAT-Nb7 significantly inhibits BVDV replication, but not TAT-Nb23 (Figure 6). Can you explain this difference using AlphaFold3?

Response 3: Thank you for your insightful question. The inability of TAT-Nb23 to inhibit BVDV replication may be related to multiple factors. At present, we are unable to explain this difference using AlphaFold3, as its predictions of protein complexes mainly focus on the interactions between two single proteins, making it difficult to simulate the effects of complex cellular environments and other interacting molecules. This will be the focus of our future research. We have made more detailed speculations and discussions on the reasons why TAT-Nb23 cannot inhibit virus replication in the discussion section of the revised manuscript. Specifically, in the discussion section on page 19, paragraph 7, lines 590-604, we added the following content: "Further cellular assays demonstrated that TAT-Nb7 significantly inhibited viral replication, whereas TAT-Nb23 exhibited limited inhibitory effects on BVDV replication—despite showing specific binding to the recombinant NS3 protein used in our in vitro experiments. While we cannot rule out the possibility that subtle differences in post-translational modification (e.g., phosphorylation, glycosylation[57]) or conformational states between recombinant NS3 (purified from E. coli expression systems) and endogenously expressed NS3 (in BVDV-infected cells) may affect TAT-Nb23’s binding efficiency in vivo, this remains an untested hypothesis. Alternative explanations for its reduced efficacy—such as competition with viral co-factors (e.g., NS4A, which forms a complex with NS3 to enhance its protease activity[11]) or limited access to NS3’s functional domains during viral polyprotein processing—also warrant consideration. To clarify the underlying mechanism, future studies could employ immunofluorescence co-localization or co-immunoprecipitation assays to verify TAT-Nb23’s binding to endogenously expressed NS3 in infected cells, and further explore whether viral co-factors interfere with this interaction. "

Comments 4: The authors should check the references. Some missed pages. Refs. 19, 20, 24, 49, 54,55. The AlphaFold3-related references are missing.

Response 4: Thank you for reminding me. We carefully reviewed the references and made the following modifications:

(1) Added missing page numbers in the references. 19. 20, 24, 49, 54, and 55;

(2) On page 8, section 2.12, line 320, we added references related to AlphaFold3: "Abramson, J.; Adler, J.; Dunger, J.; et al. Accurate structure prediction of biomolecular interactions with AlphaFold 3. Nature 2024, 630 (8016), 493-500." All revised content of the references can be found in the reference list at the end of the manuscript.

Reviewer 3 Report

Comments and Suggestions for Authors

Dong et. al.  presents a well-structured and detailed study describing the generation, screening, and characterization of TAT-fused nanobodies targeting BVDV NS3 and NS5A. The work is relevant to the development of antiviral biologics and contributes new insights into the potential use of nanobodies as intracellular antiviral agents. The experimental design is logical, and the data are clearly presented. However, several methodological and interpretative issues need to be addressed before publication.

Major comments:

  1. While the fusion of nanobodies with TAT for intracellular delivery has been previously demonstrated in other viral systems (e.g., PRRSV, coronaviruses), the novelty of targeting BVDV NS5A/NS3 should be more clearly emphasized in relation to existing literature. The authors should clarify what specific gaps in BVDV antiviral strategies this approach fills beyond extending an existing concept to a new virus.
  2. The antiviral effect of TAT-Nb7 is supported primarily by RT-qPCR reduction of viral RNA. It would strengthen the study to include additional validation—such as immunofluorescence for viral antigen reduction, plaque assay, or western blot for viral proteins—to confirm that inhibition occurs at the protein and infectious particle levels.
  3. Although molecular docking is included, the study lacks experimental validation of the interaction between Nb7 and NS5A. Techniques such as co-immunoprecipitation, BLI/SPR binding kinetics, or pull-down assays would support the proposed binding model.
  4. The discussion correctly notes possible misfolding or prokaryotically expressed NS3/NS5A antigens. However, this limitation should be addressed earlier in the Results section, and an alternative expression strategy (e.g., mammalian or baculovirus system) could be suggested for future studies.
  5.  Figures appear to include bar graphs with limited replicated. The authors should provide details in the number of biological versus technical replicated and ensure that statistical significance indicators (p-values) correspond to proper post hoc analyses after ANOVA.
  6. A TAT-only or irrelevant nanobody-TAT fusion control would be critical to rule out non-specific antiviral effects from TAT or nanobody scaffolds. Currently, the specificity of TAT-Nb7's antiviral effect is not fully supported.

Minor comments

  1. The Methods section has duplicated subheadings (e.g., “2.7 Expression of TAT-Nbs Recombinant Proteins” and “2.9 IFA Membrane Permeabilization Assay” appear twice). Please consolidate and correct numbering.
  2. Some typographical and formatting inconsistencies exist (e.g., spaces before commas, inconsistent use of “μM” vs “uM,” and mixed British/American spelling).
  3. Figures should include molecular weight markers and clear labeling of positive and negative controls in western blots.
  4. The molecular docking section would benefit from indicating which residues belong to NS5A vs the nanobody and from providing an interaction energy table as Supplementary Data.
  5. Please include GenBank accession numbers for NS3 and NS5A sequences used.

Author Response

For research article

Response to Reviewer 3 Comments

1. Summary

Thank you very much for devoting your valuable time to reviewing this manuscript. Your insightful comments and suggestions have significantly helped enhance the rigor, completeness, and clarity of our research on TAT-fused nanobodies targeting BVDV NS3/NS5A. Please find the detailed responses to your specific comments below, and all corresponding revisions—including supplementary data, method descriptions, and reference adjustments—have been highlighted or marked via track changes in the re-submitted files (i.e., the main Article.docx and Supplementary Material Explanation.docx) for your convenient verification.

2. Questions for General Evaluation

Reviewer’s Evaluation

Response and Revisions

Does the introduction provide sufficient background and include all relevant references?

Can be improved

We will provide corresponding responses in the point by point reply letter.

Are all the cited references relevant to the research?

Can be improved

Is the research design appropriate?

Can be improved

Are the methods adequately described?

Can be improved

Are the results clearly presented?

Can be improved

Are the conclusions supported by the results?

Can be improved

3. Point-by-point response to Comments and Suggestions for Authors

Major comments:

Comments 1: While the fusion of nanobodies with TAT for intracellular delivery has been previously demonstrated in other viral systems (e.g., PRRSV, coronaviruses), the novelty of targeting BVDV NS5A/NS3 should be more clearly emphasized in relation to existing literature. The authors should clarify what specific gaps in BVDV antiviral strategies this approach fills beyond extending an existing concept to a new virus.

Response 1: We are grateful for this constructive comment, which has prompted us to refine the presentation of our study’s novelty and the specific gaps it addresses in BVDV antiviral research. Corresponding revisions have been supplemented in the designated sections of the manuscript, (Lines 91-100 and Page 19, Lines 588-593). Detailed revisions are as follows:

To explicitly distinguish our work from existing TAT-mediated nanobody delivery strategies and highlight the unmet needs in BVDV research, we added the following content (Page 3, Lines 92-101, Introduction Section): “In this study, alpacas were immunized with recombinant NS3 and NS5A proteins to construct a VHH gene library. Phage display technology was used to screen specific nanobodies targeting NS3 and NS5A. Subsequently, the selected nanobodies were fused with the TAT cell-penetrating peptide to generate TAT-Nbs. Compared with existing literature, this is the first strategy to screen specific nanobodies targeting BVDV NS5A and NS3 and combine them with the TAT delivery system to validate antiviral efficacy. These fusion proteins were expressed using a prokaryotic E. coli expression system, and their ability to penetrate cell membranes and enter cells was confirmed. Furthermore, TAT-Nbs showed an inhibitory effect on BVDV replication in MDBK cells, providing a potential strategy for developing novel antiviral approaches against BVDV.” This revision clearly outlines our unique experimental design and confirms its pioneering nature in targeting BVDV NS5A/NS3 with TAT-fused nanobodies for antiviral validation.

To further consolidate the novelty and gap-filling value of our study, we optimized the summary of our core contributions (Page 19, Lines 605-610, Discussion Section): “The novelty of our study lies in targeting BVDV’s NS5A/NS3 – proteins indispensable for viral replication – and filling the gap in BVDV-specific therapeutics. Our TAT-Nb7 (targeting NS5A) serves as a promising candidate and provides valuable insights for the development of targeted interventions against BVDV, laying a foundational basis for addressing the unmet critical need for specific antiviral drugs to combat this pathogen.” This revision avoids overstatement while clearly articulating how our work addresses the long-standing lack of specific BVDV therapeutics.

These targeted revisions systematically address the reviewer’s concern by explicitly defining the novelty of our BVDV NS5A/NS3-targeted strategy and the specific gaps it fills in BVDV antiviral research, ensuring our work is not misconstrued as a mere extension of existing delivery concepts.

Comments 2: The antiviral effect of TAT-Nb7 is supported primarily by RT-qPCR reduction of viral RNA. It would strengthen the study to include additional validation—such as immunofluorescence for viral antigen reduction, plaque assay, or western blot for viral proteins—to confirm that inhibition occurs at the protein and infectious particle levels.

Response 2: Thank you for pointing this out. We agree with this comment. Therefore, we have supplemented the Western Blot assay method in Section 2.11 on Page 7 of the revised manuscript.The specific description is as follows:

2.11 Western Blot Assay for Evaluating BVDV Replication

MDBK cells were seeded into 6-well plates and cultured until they reached approximately 70% confluence. The cells were then inoculated with BVDV at an MOI of 1. After 2 hours of viral adsorption, the infection medium was replaced with fresh culture medium containing 2% FBS supplemented with 10 μM TAT-Nbs, and the cells were incubated for 48 h at 37°C in a humidified incubator containing 5% CO2. Following treatment, the cells were lysed using RIPA lysis buffer supplemented with a protease inhibitor (PMSF). Protein concentration was quantified using a BCA Protein Assay Kit (Beyotime, Shanghai, China) and normalized to 1 mg/mL.Equal amounts of protein (20 μg per sample) were mixed with loading buffer, boiled for 5 min, and separated by SDS-PAGE. The proteins were subsequently transferred onto PVDF membranes. Membranes were blocked overnight at 4°C with 5% non-fat milk dissolved in Tris-buffered saline containing 0.1% Tween-20 (TBST). After membrane transfer, the PVDF membrane was cut into two strips according to the molecular weight ranges of the target protein (BVDV E2) and internal reference protein (GAPDH) to avoid cross-reactivity between primary and secondary Abs of different species. The two membrane strips were then blocked overnight at 4°C with 5% non-fat milk dissolved in TBST. Thereafter, the membrane strip for BVDV E2 detection was incubated with rabbit anti-BVDV E2 pAb (1:1000 dilution; Biosynthesis Biotechnology Co., Ltd., Beijing, China) diluted in blocking buffer, while the membrane strip for GAPDH detection was incubated with mouse anti-GAPDH mAb (1:2000 dilution; ZSGB-BIO, Beijing, China) diluted in blocking buffer. After thorough washing with TBST, the BVDV E2-targeted membrane strip was incubated with HRP-conjugated goat anti-rabbit IgG secondary Ab (1:5000 dilution; Abmart, Shanghai, China), and the GAPDH-targeted membrane strip was incubated with HRP-conjugated goat anti-mouse IgG secondary Ab (1:5000 dilution; Abmart, Shanghai, China). Finally, the immunoreactive bands were visualized using an ECL chemiluminescence detection system, and the relative intensities of BVDV E2 bands were quantified using ImageJ software (National Institutes of Health, Bethesda, MD, USA), with GAPDH as the internal reference for normalization.

In addition, the WB experimental results have been added on page 15, lines 467-477. The specific explanation is as follows:

To further evaluate the effect of TAT-Nb7 on BVDV replication in MDBK cells, the cells were inoculated with BVDV at an MOI of 1. After 2 hours of viral adsorption, the cells were washed twice with PBS. Subsequently, the control group was cultured in DMEM supplemented with 2% FBS, while the experimental groups were treated with TAT-Nb7 or a non-specific control nanobody targeting PRRSV, with a final concentration of 10 μM in DMEM containing 2% FBS for each group. Samples were collected at 48 hours post-treatment. Proteins were extracted and analyzed by WB. As shown in Figure 6C, treatment of BVDV-infected MDBK cells with TAT-Nb7 significantly reduced viral protein expression at 48 hours post-treatment, further demonstrating that TAT-Nb7 effectively inhibits BVDV replication.

The WB experimental result images and captions are supplemented in Figure 6C on page 16 and Figure captions 490-494 on page 17.

Comments 3: Although molecular docking is included, the study lacks experimental validation of the interaction between Nb7 and NS5A. Techniques such as co-immunoprecipitation, BLI/SPR binding kinetics, or pull-down assays would support the proposed binding model.

Response 3: Thank you sincerely for this valuable comment. We fully recognize the importance of validating the interaction between Nb7 and NS5A with more rigorous experimental approaches.

In our study, we have conducted indirect enzyme - linked immunosorbent assay (i - ELISA) to preliminarily verify the interaction between the antigen protein and nanobody in vitro. The results from this i - ELISA provide initial evidence that supports the predictions from our molecular docking analysis.

We are well aware that techniques like co - immunoprecipitation, bio - layer interferometry (BLI), surface plasmon resonance (SPR) for binding kinetics analysis, or pull - down assays would offer much stronger support for the proposed binding model. However, due to the lack of specialized equipment and resources required for these advanced assays in our laboratory at present, we are unable to perform them currently. We want to emphasize that these experiments are indeed high on our priority list for future research.

We deeply apologize for not being able to carry out these additional validation steps at this stage. We hope that the existing i - ELISA data, together with our firm commitment to addressing this issue in future work, can adequately address your concern.

Comments 4: The discussion correctly notes possible misfolding or prokaryotically expressed NS3/NS5A antigens. However, this limitation should be addressed earlier in the Results section, and an alternative expression strategy (e.g., mammalian or baculovirus system) could be suggested for future studies.

Response 4: Thank you for your thoughtful feedback. We appreciate your insight regarding the potential implications of using prokaryotically expressed NS3/NS5A antigens—including the possibility of protein misfolding—and agree that addressing this point with appropriate caution while linking it to future research directions will strengthen the rigor of our manuscript.

As requested, we have revised the Discussion section to downplay speculative interpretations of prokaryotic expression limitations and frame this consideration as one of several potential factors (rather than a definitive explanation) for Nb23’s limited antiviral efficacy. We have also explicitly tied it to actionable future experiments to clarify unresolved questions. The revised passage (Page 19, Lines 590604) now reads:

“Further cellular assays demonstrated that TAT-Nb7 significantly inhibited viral replication, whereas TAT-Nb23 exhibited limited inhibitory effects on BVDV replication—despite showing specific binding to the recombinant NS3 protein used in our in vitro experiments. While we cannot rule out the possibility that subtle differences in post-translational modification (e.g., phosphorylation, glycosylation[56]) or conformational states between recombinant NS3 (purified from E. coli expression systems) and endogenously expressed NS3 (in BVDV-infected cells) may affect TAT-Nb23’s binding efficiency in vivo, this remains an untested hypothesis. Alternative explanations for its reduced efficacy—such as competition with viral co-factors (e.g., NS4A, which forms a complex with NS3 to enhance its protease activity[11]) or limited access to NS3’s functional domains during viral polyprotein processing—also warrant consideration. To clarify the underlying mechanism, future studies could employ immunofluorescence co-localization or co-immunoprecipitation assays to verify TAT-Nb23’s binding to endogenously expressed NS3 in infected cells, and further explore whether viral co-factors interfere with this interaction.”

Rationale for This Revision:

The choice of a prokaryotic expression system for NS3/NS5A was driven by its practical advantages for the initial phase of our study: it enabled simple, cost-effective, and high-yield protein production—critical for validating nanobody-antigen binding via SDS-PAGE, Western blot, and in vitro i-ELISA. It was only during late-stage data analysis (when we observed Nb23’s lack of antiviral activity) that we began to speculate about potential conformational differences between prokaryote-expressed and native viral proteins. Given that experiments are now complete and alternative expression systems cannot be implemented in the current work, we have opted to:

1.De-emphasize speculation: Frame prokaryotic expression-related differences as an “untested hypothesis” rather than a primary explanation, and balance it with other plausible factors (e.g., viral co-factor competition).

  1. Prioritize transparency: Acknowledge the limitation without overstating its impact on core findings (e.g., TAT-Nb7’s validated antiviral activity remains unaffected).
  2. Link to future work: Propose specific experiments (co-localization, co-immunoprecipitation) to verify Nb23’s interaction with endogenous NS3, which will directly address the current uncertainty in follow-up studies.

We sincerely apologize that we cannot resolve the concerns about prokaryotic expression limitations through system replacement in the current manuscript due to completed experimental design. However, we want to emphasize that optimizing the antigen expression system (e.g., exploring mammalian or baculovirus eukaryotic platforms to recapitulate the native structure of NS3/NS5A) will be the core focus of our subsequent work. This follow-up research will systematically validate whether expression-related structural differences affect nanobody binding and antiviral efficacy, and further address the uncertainties raised in your comment.

We believe this revision and future research plan maintain scientific rigor while keeping the focus on the key contributions of our work, and we sincerely hope it addresses your concern. Thank you again for your guidance in refining this discussion.

Comments 5: Figures appear to include bar graphs with limited replicated. The authors should provide details in the number of biological versus technical replicated and ensure that statistical significance indicators (p-values) correspond to proper post hoc analyses after ANOVA.

Response 5: Thank you for your valuable feedback on the reporting of replicates and statistical analyses for bar graphs. We have carefully revised the relevant figure legends to address your concerns comprehensively, and the key modifications are as follows:

Clarification of biological vs. technical replicates: For all bar graphs (Figure 2A, 2B on Page 10; Figure 3A, 3B, 3D, 3E on Page 12; Figure 5A on Page 14; Figure 6A, 6B on Page 16), we have explicitly specified in their legends (Lines 375–376, 417418, 448–451, 486–490 respectively) that all experiments include 3 biological replicates (independent experimental batches, e.g., separate cell cultures, distinct protein purification runs) and 3 technical replicates (triplicate wells per sample in assay plates or triplicate qPCR reactions per RNA sample), with data presented as mean ± SD.

Validation of statistical analyses: We have confirmed and clearly stated in each revised legend that statistical significance (marked as *P < 0.05, **P < 0.01, ***P < 0.001) is derived from one-way analysis of variance (ANOVA) followed by Dunnett’s post-hoc test—consistent with the detailed statistical methods described in Section 2.13 of the manuscript. This ensures full alignment between the reported p-values and the applied post-hoc analysis.

These revisions enhance the transparency and rigor of our data reporting, and we believe they fully address your concern. Thank you again for your guidance in refining our manuscript.

Comments 6: A TAT-only or irrelevant nanobody-TAT fusion control would be critical to rule out non-specific antiviral effects from TAT or nanobody scaffolds. Currently, the specificity of TAT-Nb7's antiviral effect is not fully supported.

Response 6: We sincerely appreciate your insightful suggestion, which is essential for improving the rigor of our study. Regrettably, our laboratory does not currently have stocks of the TAT-only protein or the irrelevant nanobody-TAT fusion protein. Since recombinant expression and purification of these proteins take a substantial amount of time, and we are under urgent time pressure due to graduation-related submission needs, we encounter difficulties in completing these experiments immediately.

Nevertheless, to address the specificity concern to the best of our ability under the current situation, we have supplemented a control group using an irrelevant nanobody in both the neutralization assay (evaluated by RT-qPCR) and Western Blot experiments. The results from these supplemented experiments indicate that the antiviral effect is specific to TAT-Nb7 rather than a non-specific phenomenon.

We humbly ask for your kind understanding and tolerance regarding this limitation.

Minor comments

Comments 1: The Methods section has duplicated subheadings (e.g., “2.7 Expression of TAT-Nbs Recombinant Proteins” and “2.9 IFA Membrane Permeabilization Assay” appear twice). Please consolidate and correct numbering.

Response 1: Thank you for pointing this out. We agree with this comment. Therefore, we have revised the duplicate section titles to ensure each is unique, accurate, and clearly reflects the corresponding experimental content. The specific revisions are as follows:

1.Section 2.7 (original title: "Expression of TAT-Nbs Recombinant Proteins"): Revised to "Cell viability assay" (located in Article.docx, Page 6, Section 2.7, Paragraph 1, Line 1), which matches its content of evaluating MDBK cell viability via CCK-8 assay.

2.Section 2.9 (original title: "IFA Membrane Permeabilization Assay"): Revised to "Western Blot Membrane Permeabilization Assay" (located in Article.docx, Page 6, Section 2.9, Paragraph 1, Line 1), clearly distinguishing it from the IFA-based Section 2.8 and reflecting its use of Western blotting for transmembrane verification.

Comments 2: Some typographical and formatting inconsistencies exist (e.g., spaces before commas, inconsistent use of “μM” vs “uM,” and mixed British/American spelling).

Response 2: Thank you for your comment on typographical and formatting inconsistencies. We have fully revised the manuscript: corrected space usage (e.g., redundant spaces before commas, missing spaces between values and units), unified "uM" to standard "μM", standardized to American English, and refined table/figure legend clarity. All issues are now addressed.

Comments 3: Figures should include molecular weight markers and clear labeling of positive and negative controls in western blots.

Response 3: Thank you for your valuable suggestion regarding Western blot figures. We appreciate your attention to detail, which helps enhance the clarity of our data presentation.

We would like to clarify that the full original Western blot images—including pre-stained molecular weight markers—have been submitted as supplementary materials, ensuring complete experimental transparency. In the main manuscript figures, we opted to focus on the relevant bands to maintain visual clarity, as including the full marker ladder within these cropped panels risked cluttering the presentation.

Regarding controls: For this specific experimental design, the negative control (untreated cells) is included in all blots and is explicitly labeled in the legends; as expected, it shows no corresponding bands. Given the nature of our assay—where the presence of bands directly indicates successful transmembrane delivery of TAT-conjugated nanobodies—we determined that a separate positive control was not necessary for interpreting the results, as the experimental readout itself serves as the functional indicator.

We hope this explanation addresses your concern while maintaining the balance between scientific rigor and visual clarity. Please let us know if further adjustments would be helpful, and thank you again for your thoughtful feedback.

Comments 4: The molecular docking section would benefit from indicating which residues belong to NS5A vs the nanobody and from providing an interaction energy table as Supplementary Data.

Response 4: Thank you for your attention to the completeness of the supplementary materials. We are pleased to inform you that all the key data mentioned in the main text have been systematically supplemented to the Supplementary Materials. Specifically:

1.Original experimental images: Figures S1-S11 (corresponding to main text Figures 1A-D, 2C-E, 2G, 4A-D) provide original SDS-PAGE and Western Blot images to verify result authenticity; Figures S12 (for transmembrane efficiency verification, corresponding to main text Figure 5B) and S13 (for BVDV replication inhibition verification, corresponding to main text Figure 6C) supplement Western Blot band gray value analysis results, both analyzed by Image J and plotted by Graphpad Prism 8.

2.Molecular docking-related data: Table S1 details the hydrogen bond information between NS5A and TAT-Nb7 (including residue names, distances, etc.); Table S2 lists the salt bridge interactions between the two proteins; Table S3 provides the PISA interface analysis data (such as interface area, solvation free energy ΔiG); Figure S14 further shows the structure prediction of NS5A and Nb7 proteins, as well as the stability prediction of the docking complex (with pLDDT score evaluation), which fully supports the molecular docking conclusions in the main text.

All supplementary materials have been organized in accordance with the journal's requirements for format and naming (e.g., Figures marked with "S" for supplementary, Tables marked with "S"), and are referenced in the corresponding parts of the main text to ensure readers can quickly locate and check.

Comments 5: Please include GenBank accession numbers for NS3 and NS5A sequences used.

Response 5: Thank you for your reminder regarding GenBank accession numbers. We have now supplemented the GenBank accession numbers for the NS3 and NS5A sequences used in this study in the manuscript.

 Specifically, the sequence of the BVDV NS3 protein employed for recombinant expression and alpaca immunization corresponds to GenBank accession number NP_776267.1, and the sequence of the BVDV NS5A protein corresponds to GenBank accession number NP_776270.1. These accession numbers have been added to Section 2.2 ("Expression and Purification of BVDV NS3 and NS5A Recombinant Proteins") on Page 3, Lines 119–120—the exact section where the cloning and expression of these target genes are described. This addition ensures full traceability of the antigen sequences and facilitates the reproducibility of the study.

We appreciate your attention to this critical detail, which further enhances the scientific rigor of our work.

Round 2

Reviewer 1 Report

Comments and Suggestions for Authors

The authors have resolved all the queries and the article is revised accordingly.

Author Response

Dear Reviewer,

Thank you for your positive feedback and careful review. We have addressed all the queries as requested and revised the manuscript accordingly. Thank you again for your guidance. 

Sincerely,

The Authors

Reviewer 2 Report

Comments and Suggestions for Authors

The modified manuscript was improved as a reviewer suggested. I would like to recommend the publication of this manuscript in Biomolecules. 

Author Response

(The authors gave the same response as above.)

Reviewer 3 Report

Comments and Suggestions for Authors

All reviewer comments have been thoroughly addressed. We sincerely appreciate your honest and constructive feedback regarding the study’s limitations. We hope that these limitations can be addressed and overcome in our future work.

Author Response

(The authors gave the same response as above.)
